# A Closer Look at the Worst-case Behavior of Multi-armed Bandit Algorithms

**Anand Kalvit[1] and Assaf Zeevi[2]**
Columbia University
New York, USA
{[1]akalvit22,[2]assaf}@gsb.columbia.edu

## Abstract

One of the key drivers of complexity in the classical (stochastic) multi-armed bandit (MAB) problem is the difference between mean rewards in the top two arms, also known as the instance gap. The celebrated Upper Confidence Bound (UCB) policy is among the simplest optimism-based MAB algorithms that naturally adapts to this gap: for a horizon of play $n$, it achieves optimal $\mathcal{O}\left(\log n\right)$ regret in instances with "large" gaps, and a near-optimal $\mathcal{O}\left(\sqrt{n \log n}\right)$ minimax regret when the gap can be arbitrarily "small." This paper provides new results on the *arm-sampling* behavior of UCB, leading to several important insights. Among these, it is shown that arm-sampling rates under UCB are asymptotically deterministic, *regardless* of the problem complexity. This discovery facilitates new sharp asymptotics and a novel alternative proof for the $\mathcal{O}\left(\sqrt{n \log n}\right)$ minimax regret of UCB. Furthermore, the paper also provides the first complete process-level characterization of the MAB problem under UCB in the conventional *diffusion scaling*. Among other things, the "small" gap worst-case lens adopted in this paper also reveals profound distinctions between the behavior of UCB and Thompson Sampling, such as an *incomplete learning* phenomenon characteristic of the latter.

## 1 Introduction

**Background and motivation.** The MAB paradigm provides a succinct abstraction of the quintessential *exploration* vs. *exploitation* trade-offs inherent in many sequential decision making problems. This has origns in clinical trial studies dating back to 1933 [30] which gave rise to the earliest known MAB heuristic, Thompson Sampling [1]. Today, the MAB problem manifests itself in various forms with applications ranging from dynamic pricing and online auctions to packet routing, scheduling, e-commerce and matching markets among others (see [9] for a comprehensive survey of different formulations). In the canonical stochastic MAB problem, a decision maker (DM) pulls one of $K$ *arms* sequentially at each time $t \in \{1, 2, ...\}$, and receives a random payoff drawn according to an arm-dependent distribution. The DM, oblivious to the statistical properties of the arms, must balance exploring new arms and exploiting the best arm played thus far in order to maximize her cumulative payoff over the horizon of play. This objective is equivalent to minimizing the *regret* relative to an oracle with perfect ex ante knowledge of the optimal arm (the one with the highest mean reward).

The classical stochastic MAB problem is fully specified by the tuple $\left(\left(\mathcal{P}_i\right)_{1 \leqslant i \leqslant K}, n\right)$, where $\mathcal{P}_i$ denotes the distribution of rewards associated with the $i^{\text{th}}$ arm, and $n$ the horizon of play.

The statistical complexity of regret minimization in the stochastic MAB problem is governed by a key primitive called the *gap*, denoted by $\Delta$, which accounts for the difference between the top two arm mean rewards in the problem. For a "well-separated" or "large gap" instance, i.e., a fixed $\Delta$ bounded away from 0, the seminal paper [22] showed that the order of the smallest achievable regret

is logarithmic in the horizon. There has been a plethora of subsequent work involving algorithms which can be fine-tuned to achieve a regret arbitrarily close to the optimal rate discovered in [22] (see [6, 14, 13, 2, 4], etc., for a few notable examples). On the other hand, no algorithm can achieve an expected regret smaller than $C\sqrt{n}$ for a fixed $n$ (the constant hides dependence on the number of arms) *uniformly* over all problem instances (also called *minimax* regret); see, e.g., [23], Chapter 15. The *saddle-point* in this minimax formulation occurs at a gap that satisfies $\Delta \asymp 1/\sqrt{n}$. This has a natural interpretation: approximately $1/\Delta^2$ samples are required to distinguish between two distributions with means separated by $\Delta$; at the $1/\sqrt{n}$-scale, it becomes statistically impossible to distinguish between samples from the top two arms within $n$ rounds of play. If the gap is smaller, despite the increased difficulty in the hypothesis test, the problem becomes "easier" from a regret perspective. Thus, $\Delta \asymp 1/\sqrt{n}$ is the statistically "hardest" scale for regret minimization. A number of popular algorithms achieve the $\sqrt{n}$ minimax-optimal rate (modulo constants), see, e.g., [4, 2], and many more do this within poly-logarithmic factors in $n$. Many of these are variations of the celebrated upper confidence bound algorithms, e.g., UCB1 [7], that achieve a minimax regret of $\mathcal{O}\left(\sqrt{n \log n}\right)$, and at the same time also deliver the logarithmic regret achievable in the instance-dependent setting of [22].

A major driver of the regret performance of an algorithm is its arm-sampling characteristics. For example, in the instance-dependent (large gap) setting, optimal regret guarantees imply that the fraction of time the optimal arm(s) are played approaches 1 in probability, as $n$ grows large. However, this fails to provide any meaningful insights as to the distribution of arm-pulls for smaller gaps, e.g., the $\Delta \asymp 1/\sqrt{n}$ "small gap" that governs the "worst-case" instance-independent setting.

**An illustrative numerical example involving "small gap."** Consider an A/B testing problem (e.g., a vaccine clinical trial) where the experimenter is faced with two competing objectives: first, to estimate the efficacy of each alternative with the best possible precision given a budget of samples, and second, keeping the overall cost of the experiment low. This is a fundamentally hard task and algorithms incurring a low cumulative cost typically spend little time exploring sub-optimal alternatives, resulting in a degraded estimation precision (see, e.g., [5]). In other words, algorithms tailored for (cumulative) regret minimization may lack *statistical power* [31]. While this trade-off is unavoidable in "well-separated" instances, numerical evidence suggests a plausible resolution in instances with "small" gaps as illustrated below. For example, such a situation might arise in trials conducted using two similarly efficacious vaccines (abstracted away as $\Delta \approx 0$). To illustrate the point more vividly, consider the case where $\Delta$ is exactly 0 (of course, this information is not known to the experimenter). This setting is numerically illustrated in Figure 1, which shows the empirical distribution of $N_1(n)/n$ (the fraction of time arm 1 is played until time $n$) in a two-armed bandit with $\Delta = 0$, under two different algorithms (UCB and Thompson Sampling), and reward configurations.

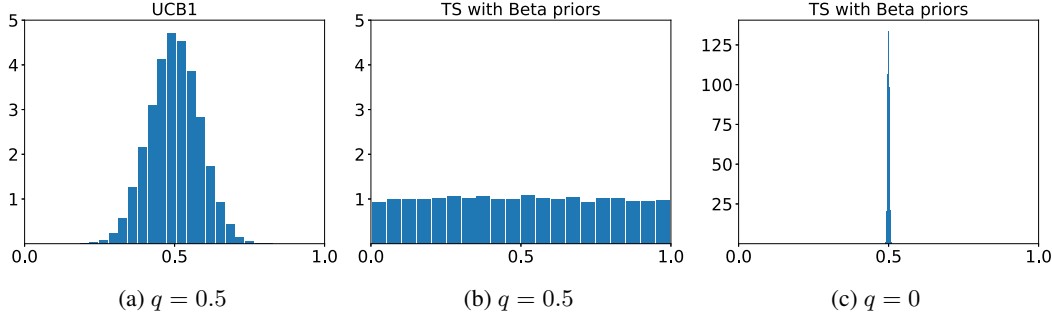

Figure 1: **Incomplete learning under Thompson Sampling.** A two-armed bandit with arms having Bernoulli($q$) rewards each: Histograms show the empirical (probability) distribution of $N_1(n)/n$ for $n = 10,000$ pulls, plotted using 20,000 experiments. Algorithms: UCB1 [7], TS with Beta priors [1].

A desirable property of the outcome in this setting is to have a *linear allocation* of the sampling budget *per arm* on almost every sample-path of the algorithm, as this leads to "complete learning:" an algorithm's ability to discern statistical indistinguishability of the arm-means, and induce a "balanced" allocation in that event. However, despite the simplicity of the zero-gap scenario, it is far from obvious whether the aforementioned property may be satisfied for standard bandit algorithms such as UCB and Thompson Sampling. Indeed, Figure 1 exhibits a striking difference between the two. The concentration around $1/2$ observable in Figure 1(a) indicates that UCB results in an approximately "balanced" sample-split, i.e., the allocation is roughly $n/2$ per arm for large $n$

(and this is observed for "most" sample-paths). In fact, we will later see that the "bell curve" in Figure 1(a) eventually collapses into the Dirac measure at $1/2$ (Theorem 1). On the other hand, under Thompson Sampling, the allocation of samples across arms *may* be arbitrarily "imbalanced" despite the arms being statistically identical, as seen in Figure 1(b) (see, for contrast, Figure 1(c), where the allocation is perfectly "balanced"). Namely, the distribution of the posterior *may* be such that arm 1 is allocated anywhere from almost no sampling effort all the way to receiving almost the entire sampling budget, as Figure 1(b) suggests. Non-degeneracy of arm-sampling rates is observable also under the more widely used version of the algorithm that is based on Gaussian priors and Gaussian likelihoods (Algorithm 2 in [2]); see Figure 2(a). Such behavior can be detrimental for ex post causal inference in the general A/B testing context, and the vaccine testing problem referenced earlier. This is demonstrated via an instructional example of a two-armed bandit with one deterministic reference arm (aka the "one-armed" bandit paradigm), illustrated in Figure 2, and discussed below.

**A numerical example illustrating inference implications.** Consider a model where arm 1 returns a constant reward of $0.5$, while arm 2 yields rewards distributed as Bernoulli$(0.5)$. In this setup, the estimate of the gap $\Delta$ (*average treatment effect* in causal inference parlance) after $n$ rounds of play is given by $\hat{\Delta} = \bar{X}_2(n) - 0.5$, where $\bar{X}_2(n)$ denotes the empirical mean reward of arm 2 at time $n$. The $\mathcal{Z}$ statistic associated with this gap estimator is given by $\mathcal{Z} = 2\sqrt{N_2(n)}\hat{\Delta}$, where $N_2(n)$ is the visitation count of arm 2 at time $n$. In the absence of any sample-adaptivity in the arm 2 data, results from classical statistics such as the Central Limit Theorem (CLT) would posit an asymptotically Normal distribution for $\mathcal{Z}$. However, since the algorithms that play the arms are adaptive in nature, e.g., UCB and Thompson Sampling, asymptotic-normality may no longer be guaranteed. Indeed, the numerical evidence in Figure 2(b) strongly points to a significant departure from asymptotic-normality of the $\mathcal{Z}$ statistic associated with the gap estimator under Thompson Sampling (TS). Non-normality of the $\mathcal{Z}$ statistic can be problematic for inferential tasks, e.g., it can lead to statistically unsupported inferences in the binary hypothesis test $\mathcal{H}_0 : \Delta = 0$ vs. $\mathcal{H}_1 : \Delta \neq 0$ performed using confidence intervals constructed as per the conventional CLT approximation. In sharp contrast, our work shows that UCB satisfies a certain "balanced" sampling property (such as that in Figure 1(a)) in instances with "small" gaps, formally stated as Theorem 1, that drives the $\mathcal{Z}$ statistic towards asymptotic-normality in the aforementioned binary hypothesis testing example (asymptotic-normality being a consequence of Theorem 4). Furthermore, since the $\sqrt{n}$-normalized "stochastic" regret (defined in (1) in §2) equals $-\left(\sqrt{N_2(n)/(4n)}\right)\mathcal{Z}$, it follows that this too, satisfies asymptotic-normality under UCB (due to Theorem 4, in conjunction with Theorem 1). These properties are evident in Figure 2(c) below, and signal reliability of ex post causal inference (under classical assumptions like validity of CLT) from "small gap" data collected by UCB vis-à-vis Thompson Sampling (TS). The reliability of inference under TS may be doubtful even in the limit of infinite data, as Figure 2(b) suggests.

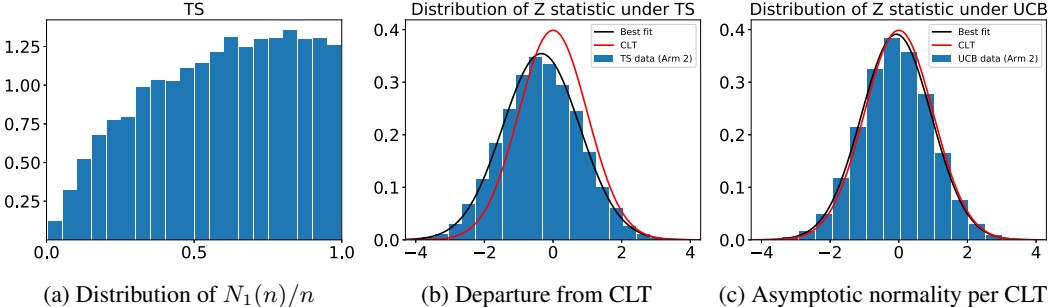

(a) Distribution of $N_1(n)/n$     (b) Departure from CLT     (c) Asymptotic normality per CLT

Figure 2: **Failure of CLT under TS.** A two-armed bandit with $\Delta = 0$: Arm 1 returns a constant reward of $0.5$, and arm 2 yields rewards distributed as Bernoulli$(0.5)$. In (a), the histogram shows the empirical (probability) distribution of $N_1(n)/n$. Algorithms: TS (Algorithm 2 in [2]) and UCB (UCB1 in [7]). All histograms have $n = 10,000$ pulls, and are plotted using $\aleph = 20,000$ experiments.

While traditional literature on stochastic bandits is dedicated primarily to the regret minimization problem, there has been significant recent interest also in finer-grain properties of popular "adaptive" MAB algorithms such as UCB and Thompson Sampling. For example, a recent line of work ([27, 28, 29]) investigates the "bias" of optimistic algorithms like UCB. The focus of our work is on understanding the distribution of arm-pulls, which as discussed earlier, has significant bearings on

ex post causal inference from data collected adaptively by bandit algorithms (see, e.g., [33, 16, 32], etc., and references therein for recent developments), algorithmic fairness in the broader context of fairness in machine learning (see [25] for a survey), as well as on novel formulations of the MAB problem such as [20]. Below, we discuss extant literature relevant to our line of work.

**Previous work.** The study of "well-separated" instances, or the *large gap* regime, is supported by rich literature. For example, [6] provides high-probability bounds on arm-sampling rates under a parametric family of UCB algorithms. However, as the gap diminishes, leading to the so called *small gap* regime, the aforementioned bounds become vacuous. The understanding of arm-sampling behavior remains relatively under-studied here even for popular algorithms such as UCB and Thompson Sampling. This regime is of special interest in that it also covers the classical *diffusion scaling*[1], where $\Delta \asymp 1/\sqrt{n}$, which as discussed earlier, corresponds to instances that statistically constitute the "worst-case" for hypothesis testing and regret minimization. Recently, a partial diffusion-limit characterization of the arm-sampling distribution under a version of Thompson Sampling with horizon-dependent prior variances[2], was provided in [32] as a solution to a certain stochastic differential equation (SDE). The numerical solution to said SDE was observed to have a non-degenerate distribution on $[0, 1]$. Similar numerical observations on non-degeneracy of the arm-sampling distribution also under standard versions of Thompson Sampling were reported in [10, 20], among others, albeit limited only to the special case of $\Delta = 0$, and absent a theoretical explanation for the aforementioned observations. Thus, outside of the so called "easy" problems, where $\Delta$ is bounded away from 0 by an absolute constant, theoretical understanding of the sampling behavior of bandit algorithms remains an open area of research.

**Contributions.** In this paper, we provide the first complete asymptotic characterization of arm-sampling distributions under canonical UCB (Algorithm 1) as a function of the gap $\Delta$ (Theorem 1). This gives rise to a fundamental insight: arm-sampling rates are asymptotically deterministic under UCB regardless of the hardness of the instance. We also provide the first theoretical explanation for an "incomplete learning" phenomenon under Thompson Sampling (Algorithm 2) alluded to in Figure 1, as well as a sharp dichotomy between Thompson Sampling and UCB evident therein (Theorem 2). This result earmarks an "instability" of Thompson Sampling in terms of the limiting arm-sampling distribution. As a sequel to Theorem 1, we provide the first *complete* characterization of the worst-case performance of canonical UCB (Theorem 3). One consequence is that the $\mathcal{O}\left(\sqrt{n \log n}\right)$ minimax regret of UCB is strictly unimprovable in a precise sense. Moreover, our work also leads to the first process-level characterization of the two-armed bandit problem under canonical UCB in the classical *diffusion limit*, according to which a suitably normalized cumulative reward process converges in law to a Brownian motion with fully characterized drift and infinitesimal variance (Theorem 4). To the best of our knowledge, this is the first such characterization of UCB-type algorithms. Theorem 4 facilitates a complete distribution-level characterization of UCB's diffusion-limit regret, thereby providing sharp insights as to the problem's minimax complexity. Such distribution-level information may also be useful for a variety of inferential tasks, e.g., construction of confidence intervals (see the binary hypothesis testing example referenced in Figure 2(c)), among others. We believe our results may also present new design considerations, in particular, how to achieve, loosely speaking, the "best of both worlds" for Thompson Sampling, by addressing its "small gap" instability. Lastly, we note that our proof techniques are markedly different from the conventional methodology adopted in MAB literature ([6, 9, 2]), and may be of independent interest in the study of related learning algorithms.

**Organization of the paper.** A formal description of the model and the canonical UCB algorithm is provided in §2. All theoretical propositions are stated in §3, along with a high-level overview of their scope and proof sketch; detailed proofs and ancillary results are relegated to the supplementary material. Finally, concluding remarks and open problems are presented in §4.

## 2 The model and notation

The technical development in this paper will focus on the two-armed problem purely for expositional reasons; we provide extensions to the general $K$-armed setting in Appendix B. The restriction to two-armed bandits has precedence also in the literature due to its tractability for sharp asymptotic

---

[1]This is a standard technique for performance evaluation of stochastic systems, commonly used in the operations research and mathematics literature, see, e.g., [15].

[2]Assumed to be vanishing in $n$; standard versions of the algorithm involve fixed (positive) prior variances.

analyses, see, e.g., [21]. This setting encapsulates the core statistical complexity of the MAB problem in the "small gap" regime, as well as concisely highlighting the key novelties in our approach. Before describing the model formally, we introduce the following asymptotic conventions.

**Notation.** We say $f(n) = o(g(n))$ or $g(n) = \omega(f(n))$ if $\lim_{n\to\infty} \frac{f(n)}{g(n)} = 0$. Similarly, $f(n) = \mathcal{O}(g(n))$ or $g(n) = \Omega(f(n))$ if $\limsup_{n\to\infty} \left|\frac{f(n)}{g(n)}\right| \leqslant C$ for some constant $C$. If $f(n) = \mathcal{O}((g(n)))$ and $f(n) = \Omega((g(n)))$ hold simultaneously, we say $f(n) = \Theta(g(n))$, or $f(n) \asymp g(n)$, and we write $f(n) \sim g(n)$ in the special case where $\lim_{n\to\infty} \frac{f(n)}{g(n)} = 1$. If either sequence $f(n)$ or $g(n)$ is random, and one of the aforementioned ratio conditions holds in probability, we use the subscript $p$ with the corresponding Landau symbol. For example, $f(n) = o_p(g(n))$ if $f(n)/g(n) \xrightarrow{p} 0$ as $n \to \infty$. Lastly, the notation '$\Rightarrow$' will be used for weak convergence.

**The model.** The arms are indexed by $\{1, 2\}$. Each arm $i \in \{1, 2\}$ is characterized by a reward distribution $\mathcal{P}_i$ supported on $[0, 1]$ with mean $\mu_i$. The difference between the two mean rewards, aka the *gap*, is given by $\Delta = |\mu_1 - \mu_2|$; as discussed earlier, this captures the *hardness* of an instance. The sequence of rewards associated with the *first $m$ pulls* of arm $i$ is denoted by $(X_{i,j})_{1 \leqslant j \leqslant m}$. The rewards are assumed to be i.i.d. in time, and independent across arms.[3] The number of pulls of arm $i$ up to (and including) time $t$ is denoted by $N_i(t)$. A policy $\pi := (\pi_t)_{t \in \mathbb{N}}$ is an adapted sequence that prescribes pulling an arm $\pi_t \in \mathcal{S}$ at time $t$, where $\mathcal{S}$ denotes the probability simplex on $\{1, 2\}$. The natural filtration at time $t$ is given by $\mathcal{F}_t := \sigma\left\{(\pi_s)_{s \leqslant t}, \left((X_{i,j})_{j \leqslant N_i(t)} : i = 1, 2\right)\right\}$. The stochastic regret of policy $\pi$ after $n$ plays, denoted by $R_n^\pi$, is given by

$$R_n^\pi := \sum_{t=1}^{n} \left[\max(\mu_1, \mu_2) - X_{\pi_t, N_{\pi_t}(t)}\right]. \tag{1}$$

The decision maker is interested in the problem of minimizing the *expected regret*, given by

$$\inf_{\pi \in \Pi} \mathbb{E} R_n^\pi,$$

where $\Pi$ is the set of policies satisfying the non-anticipation property $\pi_t : \mathcal{F}_{t-1} \to \mathcal{S}$, $1 \leqslant t \leqslant n$, and the expectation is w.r.t. the randomness in reward realizations as well as possible randomness in the policy $\pi$. In this paper, we will focus primarily on the canonical UCB policy given by Algorithm 1 below. This policy is parameterized by an exploration coefficient $\rho$, which controls its arm-exploring rate. The standard UCB1 policy [7] corresponds to Algorithm 1 with $\rho = 2$; the effect of $\rho$ on the expected and high-probability regret bounds of the algorithm is well-documented in [6] for problems with a "large gap." In what follows, $\bar{X}_i(t-1)$ denotes the empirical mean reward from arm $i \in \{1, 2\}$ at time $t - 1$, i.e., $\bar{X}_i(t-1) := \frac{\sum_{j=1}^{N_i(t-1)} X_{i,j}}{N_i(t-1)}$.

---

**Algorithm 1** The canonical UCB policy for two-armed bandits.

---

1: **Input:** Exploration coefficient $\rho \in \mathbb{R}_+$.
2: At $t = 1, 2$, play each arm $i \in \{1, 2\}$ once.
3: **for** $t \in \{3, 4, ...\}$ **do**
4:     Play arm $\pi_t \in \arg\max_{i \in \{1,2\}} \left(\bar{X}_i(t-1) + \sqrt{\frac{\rho \log(t-1)}{N_i(t-1)}}\right)$.

---

## 3 Main results

Algorithm 1 is known to achieve $\mathbb{E} R_n^\pi = \mathcal{O}(\log n)$ in the instance-dependent setting, and $\mathbb{E} R_n^\pi = \mathcal{O}(\sqrt{n \log n})$ in the "small gap" minimax setting. The primary focus of this paper is on the distribution of arm-sampling rates, i.e., $N_i(n)/n$, $i \in \{1, 2\}$. Our main results are split across two sub-sections; §3.1 examines the behavior of UCB (Algorithm 1) as well as another popular bandit algorithm, Thompson Sampling (specified in Algorithm 2). §3.2 is dedicated to results on the (stochastic) regret of Algorithm 1 under the $\Delta \asymp \sqrt{(\log n)/n}$ "worst-case" gap and the $\Delta \asymp 1/\sqrt{n}$ "diffusion-scaled" gap.

---

[3]These assumptions can be relaxed in the spirit of [7]; our results also extend to sub-Gaussian rewards.

### 3.1 Asymptotics of arm-sampling rates

**Theorem 1 (Arm-sampling rates under UCB)** *Let $i^* \in \arg\max\{\mu_i : i = 1, 2\}$ with ties broken arbitrarily. Then, the following results hold for arm $i^*$ as $n \to \infty$ under Algorithm 1 with $\rho > 1$:*

*(I) "Large gap:" If $\Delta = \omega\left(\sqrt{\frac{\log n}{n}}\right)$, then $N_{i^*}(n)/n \xrightarrow{p} 1$.*

*(II) "Small gap:" If $\Delta = o\left(\sqrt{\frac{\log n}{n}}\right)$, then $N_{i^*}(n)/n \xrightarrow{p} 1/2$.*

*(III) "Moderate gap:" If $\Delta \sim \sqrt{\frac{\theta \log n}{n}}$ for some fixed $\theta \geqslant 0$, then $N_{i^*}(n)/n \xrightarrow{p} \lambda_\rho^*(\theta)$, where the limit is the unique solution (in $\lambda$) to*

$$\frac{1}{\sqrt{1 - \lambda}} - \frac{1}{\sqrt{\lambda}} = \sqrt{\frac{\theta}{\rho}}, \tag{2}$$

*and is monotone increasing in $\theta$, with $\lambda_\rho^*(0) = 1/2$ and $\lambda_\rho^*(\theta) \to 1$ as $\theta \to \infty$.*

**Remark 1 (Permissible values of $\rho$ in Algorithm 1)** *For $\rho > 1$, the expected regret of the policy $\pi$ given by Algorithm 1 is bounded as $\mathbb{E}R_n^\pi \leqslant C\rho\left(\frac{\log n}{\Delta} + \frac{\Delta}{\rho - 1}\right)$ for some absolute constant $C > 0$; the upper bound becomes vacuous for $\rho \leqslant 1$ (see [6], Theorem 7). We therefore restrict Theorem 1 to $\rho > 1$ to ensure that $\mathbb{E}R_n^\pi$ remains non-trivially bounded for all $\Delta$.*

**Discussion and intuition.** Theorem 1 essentially asserts that the sampling rates $N_i(n)/n$, $i \in \{1, 2\}$ are asymptotically deterministic in probability under canonical UCB; $\Delta$ only serves to determine the value of the limiting constant. The "moderate" gap regime offers a continuous interpolation from instances with zero gaps to instances with "large" gaps as $\theta$ sweeps over $\mathbb{R}_+$ in that $\lambda_\rho^*(\theta)$ increases monotonically from $1/2$ at $\theta = 0$ to $1$ at $\theta = \infty$, consistent with intuition. The special case of $\theta = 0$ is numerically illustrated in Figure 1(a). The tails of $N_{i^*}(n)/n$ decay polynomially fast near the end points of the interval $[0, 1]$ with the best possible rate approaching $\mathcal{O}\left(n^{-3}\right)$, occurring for $\theta = 0$. However, as $N_{i^*}(n)/n$ approaches its limit, convergence becomes slower and is dominated by fatter $\Theta\left(\sqrt{\frac{\log \log n}{\log n}}\right)$ tails. The behavior in this regime is regulated by the $\mathcal{O}\left(\sqrt{n \log \log n}\right)$ envelope of the zero-drift random walk process driving the algorithm's regret (see proof of Theorem 1 for details).

**Proof sketch.** To provide the most intuitive explanation, we pivot to the special case where the arms have *identical* reward distributions, and in particular, $\Delta = 0$. The natural candidate then for the limit of the empirical sampling rate is $1/2$. On a high level, the proof relies on polynomially decaying bounds in $n$ for $\epsilon$-deviations of the form $\mathbb{P}\left(\left|\frac{N_1(n)}{n} - \frac{1}{2}\right| \geqslant \epsilon\right)$ derived using the standard trick for bounding the number of pulls of any arm on a given sample-path, to wit, for any $u, n \in \mathbb{N}$, $N_1(n)$ can be bounded above by $u + \sum_{t=u+1}^{n} \mathbb{1}\{\pi_t = 1, N_1(t-1) \geqslant u\}$, *path-wise*. Setting $u = \lceil(1/2 + \epsilon)n\rceil$ in this expression, one can subsequently show via an analysis involving careful use of the policy structure together with appropriate Chernoff bounds that with high probability (approaching 1 as $n \to \infty$), $N_1(n)/n \leqslant 1/2 + \varepsilon_\rho$ for some $\varepsilon_\rho \in (0, 1/2)$ that depends only on $\rho$. An identical result would naturally hold also for the other arm by symmetry arguments, and therefore we arrive at a meta-conclusion that $N_i(n)/n \geqslant 1/2 - \varepsilon_\rho > 0$ for both arms $i \in \{1, 2\}$ with high probability (approaching 1 as $n \to \infty$). It is noteworthy that said conclusion cannot be arrived at for an arbitrary $\epsilon > 0$ (in place of $\varepsilon_\rho$) since the polynomial upper bounds on $\mathbb{P}\left(\left|\frac{N_1(n)}{n} - \frac{1}{2}\right| \geqslant \epsilon\right)$ derived using the aforementioned path-wise upper bound on $N_1(n)$, become vacuous if $u$ is set "too close" to $n/2$, i.e., if $\epsilon$ is "near" 0. Extension to the full generality of $\epsilon > 0$ is achieved via a refined analysis that uses the Law of the Iterated Logarithm (see [11], Theorem 8.5.2), together with the previous meta-conclusion, to obtain fatter $\mathcal{O}\left(\sqrt{\frac{\log \log n}{\log n}}\right)$ tail bounds when $\epsilon$ is near 0. Here, it is imperative to point out that the "$\log n$" appearing in the denominator is essentially from the $\sqrt{\rho \log t}$ optimistic bias term of UCB (see Algorithm 1), and therefore the convergence will, as such, hold also for other variations of the policy that have "less aggressive" $\omega(\log \log t)$ exploration functions vis-à-vis $\log t$. However, this will be achieved at the expense of the policy's expected regret performance,

as noted in Remark 1. We also note that the extremely slow $\mathcal{O}\left(\sqrt{\frac{\log\log n}{\log n}}\right)$ convergence is not an artifact of our analysis, but in fact, supported by the numerical evidence in Figure 1(a), suggestive of a plausible non-convergence (to $1/2$) in the limit. We believe such observations in previous works likely led to incorrect folk conjectures ruling out the existence of a deterministic limit under UCB à la Theorem 1 (see, e.g., [10] and references therein). The proof for a general $\Delta$ in the "small" and "moderate" gap regimes is skeletally similar to that for $\Delta = 0$, albeit guessing a candidate limit for $N_{i^*}(n)/n$ is non-trivial; a closed-form expression for $\lambda_\rho^*(\theta)$ is provided in Appendix A. Full details of the proof of Theorem 1 are provided in Appendix D,E,F. $\qquad\qquad\square$

**Remark 2 (Possible generalizations of Theorem 1)** *A simple extension to the $K$-armed setting is provided in Appendix B as Theorem 5. The behavior of UCB policies is largely governed by their optimistic bias. While Theorem 1 only covers the generic UCB policy with $\sqrt{\rho \log t}$ bias, results of the form $N_i(n)/n \xrightarrow{p} c_i$ for some $c_i \in (0, 1)$ continue to hold also under smaller $\omega\left(\sqrt{\rho \log \log t}\right)$ bias (driven by the Law of the Iterated Logarithm). We believe this observation will be useful when examining more complicated UCB-inspired policies such as KL-UCB [13], DMED [18], etc.*

**What about Thompson Sampling?** Results such as those discussed above for other popular adaptive algorithms like Thompson Sampling[4] are only arable in "well-separated" instances where $N_{i^*}(n)/n \xrightarrow{p} 1$ as $n \to \infty$ follows as a trivial consequence of its $\mathcal{O}\left(\sqrt{n}\right)$ minimax regret bound [2]. For smaller gaps, theoretical understanding of the distribution of arm-pulls under Thompson Sampling remains largely absent even for its most widely-studied variants. In this paper, we provide a first result in this direction: Theorem 2 formalizes a revealing observation for classical Thompson Sampling (Algorithm 2) in instances with zero gap, and elucidates its instability in view of the numerical evidence reported in Figure 1(b) and 1(c). This result also offers an explanation for the sharp contrast with the statistical behavior of canonical UCB (Algorithm 1) à la Theorem 1, also evident from Figure 1(a). In what follows, rewards are assumed to be Bernoulli, and $S_i$ (respectively $F_i$) counts the number of successes/1's (respectively failures/0's) associated with arm $i \in \{1, 2\}$.

---

**Algorithm 2** Thompson Sampling for the two-armed Bernoulli bandit.

---

1: **Initialize:** Number of successes (1's) and failures (0's) for each arm $i \in \{1, 2\}$, $(S_i, F_i) = (0, 0)$.
2: **for** $t \in \{1, 2, ...\}$ **do**
3:     Sample for each $i \in \{1, 2\}$, $\mathcal{T}_i \sim \text{Beta}(S_i + 1, F_i + 1)$.
4:     Play arm $\pi_t \in \arg\max_{i \in \{1,2\}} \mathcal{T}_i$ and observe reward $r_t \in \{0, 1\}$.
5:     Update success-failure counts: $S_{\pi_t} \leftarrow S_{\pi_t} + r_t$, $F_{\pi_t} \leftarrow F_{\pi_t} + 1 - r_t$.

---

**Theorem 2 (Incomplete learning under Thompson Sampling)** *In a two-armed model where both arms yield rewards distributed as Bernoulli$(q)$, the following holds under Algorithm 2 as $n \to \infty$:*

*(I) If $q = 0$, then $N_1(n)/n \Rightarrow 1/2$.*

*(II) If $q = 1$, then $N_1(n)/n \Rightarrow$ Uniform distribution on $[0, 1]$.*

**Proof sketch.** The proof of Theorem 2 relies on a careful application of two subtle properties of the Beta distribution (Fact 2 and Fact 3), stated and proved in Appendix C,K. For part (I), we invoke symmetry to deduce $\mathbb{E}N_1(n) = n/2$, and use Fact 2 to show that the standard deviation of $N_1(n)$ is sub-linear in $n$, thus proving the stated assertion in (I). More elaborately, Fact 2 states for the reward configuration in (I) that the probability of playing arm 1 after it has already been played $n_1$ times, and arm 2 $n_2$ times, equals $(n_2 + 1)/(n_1 + n_2 + 2)$. This probability is smaller than $1/2$ if $n_1 > n_2$, which provides an intuitive explanation for the fast convergence of $N_1(n)/n$ to $1/2$ observed in Figure 1(c). In fact, we conjecture that the result in (I) holds also with probability 1 based on the aforementioned "self-balancing" property. The conclusion in part (II) hinges on an application of Fact 3 to show the *stronger* result: $N_1(n)$ is *uniformly distributed* over $\{0, 1, ..., n\}$ for any $n \in \mathbb{N}$. Contrary to Fact 2, Fact 3 states that quite the opposite is true for the reward configuration in (II): the probability of playing arm 1 after it has already been played $n_1$ times, and arm 2 $n_2$ times, equals $(n_1 + 1)/(n_1 + n_2 + 2)$, which is greater than $1/2$ when $n_1 > n_2$. That is, the posterior distributions

---

[4]This is the version based on Gaussian priors and Gaussian likelihoods, not the classical version based on Beta priors and Bernoulli likelihoods which has a minimax regret of $\mathcal{O}\left(\sqrt{n \log n}\right)$ [2].

evolve in such a way that the algorithm is "deceived" into incorrectly believing one of the arms (arm 2 in this case) to be inferior. This leads to large sojourn times between successive visitations of arm 2 on such a sample-path, thereby resulting in a perpetual "imbalance" in the sample-counts. This provides an intuitive explanation for the non-degeneracy observed in Figure 1(b) and 2(a), which additionally, also indicates that such behavior, in fact, persists also for general (non-deterministic) reward distributions, as well as under the Gaussian prior-based version of the algorithm. Full proof of Theorem 2 is provided in Appendix G. □

**More on "incomplete learning."** The zero-gap setting is a special case of the "small gap" regime where canonical UCB guarantees a $(1/2, 1/2)$ sample-split in probability (Theorem 1). On the other hand, Theorem 2 suggests that second order factors such as the mean signal strength (magnitude of the mean reward) could significantly affect the nature of the resulting sample-split under Thompson Sampling. Note that even though the result only presupposes deterministic 0/1 rewards, the aforementioned claim is, in fact, borne out by the numerical evidence in Figure 1(b) and 1(c). The sampling distribution seemingly flattens rapidly from the Dirac measure at $1/2$ to the Uniform distribution on $[0, 1]$ as the mean rewards move away from 0. This uncertainty in the limiting sampling behavior has non-trivial implications for a variety of application areas of such learning algorithms. For instance, a uniform distribution of arm-sampling rates on $[0, 1]$ indicates that the sample-split could be arbitrarily imbalanced along a sample-path, despite, as in the setting of Theorem 2, the two arms being statistically identical; this phenomenon is typically referred to as "incomplete learning" (see [26, 24] for the original context). Non-degeneracy in the limiting distribution is also observable numerically up to diffusion-scale gaps of $\mathcal{O}\left(1/\sqrt{n}\right)$ under other versions of Thompson Sampling (see [32] for examples); our focus on the more extreme zero-gap setting simplifies the illustration of these effects.

**A brief survey of Thompson Sampling.** While extant literature does not provide any explicit result for Thompson Sampling characterizing its arm-sampling behavior in instances with "small" and "moderate" gaps, there has been recent work on its analysis in the $\Delta \asymp 1/\sqrt{n}$ regime under what is known as the *diffusion approximation* lens (see [32, 12]). Cited works study Thompson Sampling primarily under the assumption that the prior variance associated with the mean reward of any arm vanishes in the horizon of play at an "appropriate" rate.[5] Such a scaling, however, is not ideal from a regret standpoint and indeed, the versions of Thompson Sampling optimized for regret performance use fixed (non-vanishing) prior variances, e.g., Algorithm 2 and its Gaussian prior-based counterpart (see [2]). On a high level, [32, 12] establish that as $n \to \infty$, the pre-limit $\left(N_i(nt)/n\right)_{t \in [0,1]}$ under Thompson Sampling converges weakly to a "diffusion-limit" stochastic process on $t \in [0, 1]$. Recall from earlier discussion that $\Delta \asymp 1/\sqrt{n}$ is covered under the "small gap" regime; consequently, it follows from Theorem 1 that the analogous limit for UCB is, in fact, the deterministic process $t/2$. In sharp contrast, the diffusion-limit process under Thompson Sampling may at best be characterizable only as a solution (possibly non-unique) to an appropriate stochastic differential equation or ordinary differential equation driven by a suitably (random) time-changed Brownian motion. Consequently, the diffusion limit under Thompson Sampling is more difficult to interpret vis-à-vis UCB (see Theorem 4), and it is much harder to obtain lucid insights as to the nature of the distribution of $N_i(n)/n$ as $n \to \infty$. The asymptotic distribution of $N_i(n)/n$ under Thompson Sampling is also investigated in [19], albeit in a significantly different setting. Cited paper considers the Bayesian setting where a prior distribution exists over problem instances, and the Thompson Sampling algorithm is "well-specified," i.e., information about said prior is baked into the algorithm. Specifically, a sample path of the algorithm in their model involves a *random* problem instance from the instance-space. In contrast, the derivation of the asymptotic distribution of arm-pulls in our work is for specific (fixed) problem instances, viz., reward configurations (I) and (II) described in Theorem 2, and under the classical version of Thompson Sampling (Algorithm 2).

### 3.2   Beyond arm-sampling rates

This part of the paper is dedicated to a more fine-grained analysis of the "stochastic" regret of UCB (defined in (1) in §2). Results are largely facilitated by insights on the sampling behavior of UCB in instances with "small" gaps, attributable to Theorem 1; however, we believe they are of interest in their own right. We commence with an application of Theorem 1 which provides the first complete characterization of the worst-case (minimax) performance of canonical UCB. A full diffusion-limit characterization of the two-armed bandit problem under UCB is provided thereafter in Theorem 4.

---

[5]The only result applicable to the case of non-vanishing prior variances is Theorem 4.2 of [12].

**Theorem 3 (Minimax regret complexity of UCB)** *In the "moderate gap" regime referenced in Theorem 1 where $\Delta \sim \sqrt{\frac{\theta \log n}{n}}$, the regret of the policy $\pi$ given by Algorithm 1 with $\rho > 1$ satisfies*

$$\frac{R_n^\pi}{\sqrt{n \log n}} \Rightarrow \sqrt{\theta}\left(1 - \lambda_\rho^*(\theta)\right) =: h_\rho(\theta) \quad as \ n \to \infty, \tag{3}$$

*where $\lambda_\rho^*(\theta)$ is the (unique) solution to (2).*

To the best of our knowledge, this is the first *algorithm-specific* result (sharp asymptotic) that is distinct from the general $\Omega\left(\sqrt{n}\right)$ information-theoretic lower bound by a horizon-dependent factor.[6]

**Discussion.** A closed-form expression for $\lambda_\rho^*(\theta)$ and $h_\rho(\theta)$ is provided in Appendix A. The behavior of $h_\rho(\theta)$ is illustrated below in Figure 3. For a fixed $\rho$, the function $h_\rho(\theta)$ is numerically observed to be uni-modal in $\theta$ and admit a global maximum at a unique $\theta_\rho^* := \arg\sup_{\theta \geqslant 0} h_\rho(\theta)$, bounded away from 0. Theorem 3 establishes that the worst-case (instance-independent) regret admits the sharp asymptotic $R_n^\pi \sim h_\rho\left(\theta_\rho^*\right)\sqrt{n \log n}$. In standard bandit parlance, this substantiates that the

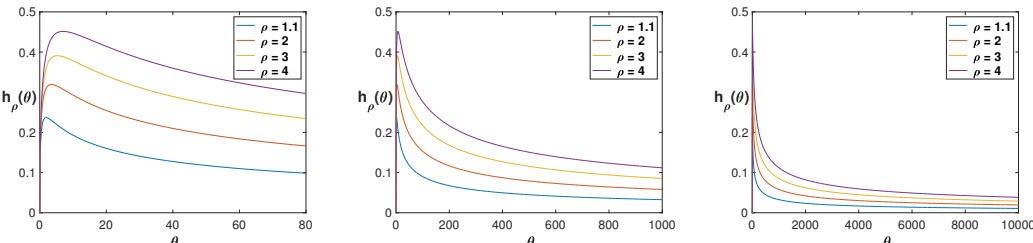

Figure 3: $h_\rho(\theta)$ **vs.** $\theta$ for different values of the exploration coefficient $\rho$ in Algorithm 1. The graphs exhibit a unique global maximizer $\theta_\rho^*$ for each $\rho$. The $\left(\theta_\rho^*, h_\rho\left(\theta_\rho^*\right)\right)$ pairs for $\rho \in \{1.1, 2, 3, 4\}$ in that order are: $(1.9, 0.2367), (3.5, 0.3192), (5.3, 0.3909), (7, 0.4514)$.

$\mathcal{O}\left(\sqrt{n \log n}\right)$ worst-case (minimax) performance guarantee of canonical UCB cannot be improved in terms of its horizon-dependence. In addition, the result also specifies the precise asymptotic constants achievable in the worst-case setting. This can alternately be viewed as a direct approach to proving the $\mathcal{O}\left(\sqrt{n \log n}\right)$ performance bound for UCB vis-à-vis conventional minimax analyses such as [9].

**Proof sketch.** On a high level, note that when $\Delta = \sqrt{(\theta \log n)/n}$, it follows from Theorem 1 that $\mathbb{E}R_n^\pi = \sqrt{(\theta \log n)/n}\mathbb{E}\left[n - N_{i^*}(n)\right] \sim h_\rho(\theta)\sqrt{n \log n}$ (convergence in probability implies that in mean since $|N_{i^*}(n)/n| \leqslant 1$). That $R_n^\pi$ also admits the same sharp asymptotic can be shown via a finer analysis. In other regimes of $\Delta$, viz., "small" and "large" gaps, we already know that $R_n^\pi = o_p\left(\sqrt{n \log n}\right)$. This is obvious for "small" $\Delta$ since $\mathbb{E}R_n^\pi \leqslant \Delta n = o\left(\sqrt{n \log n}\right)$, while for "large" $\Delta$, we use $\mathbb{E}R_n^\pi \leqslant C\rho\left((\log n)/\Delta + 1/(\rho-1)\right)$ for some absolute constant $C$ (given $\rho > 1$, $\Delta \leqslant 1$) [6], followed by Markov's inequality. Thus, the multiplicative constant $\sup_{\theta \geqslant 0} h_\rho(\theta)$ obtained in the "moderate" gap regime must correspond to the worst-case performance of the algorithm. $\square$

**Towards diffusion asymptotics.** Diffusion scaling is a classical stochastic analysis tecnnique widely used in the mathematics and operations research literature, see, e.g., steady-state analyses of queuing systems in [15], and a recent application to certain sequential testing problems in [3]. Under this lens, time is accelerated linearly in $n$, space contracted by a factor of $\sqrt{n}$, and a sequence of systems indexed by $n$ is considered. In our problem, the $n^{\text{th}}$ such system refers to an instance of the two-armed bandit with: $n$ as the horizon of play; a gap that vanishes in the horizon as $\Delta = c/\sqrt{n}$ for some fixed $c$; and fixed reward variances given by $\sigma_1^2, \sigma_2^2$. This is a natural scaling for MAB experiments in that it "preserves" the hardness of the learning problem as $n$ sweeps over the sequence of systems. Recall also from previous discussion that the "hardest" information-theoretic instances have a $\Theta\left(1/\sqrt{n}\right)$ gap; in short, the diffusion limit is an appropriate asymptotic lens for observing interesting process-level behavior in the MAB problem. However, despite the aforementioned reasons, the diffusion limit behavior of bandit algorithms remains poorly understood and largely unexplored. A recent foray

---

[6]Previous work establishes matching $\Theta\left(\sqrt{Kn \log K}\right)$ upper and lower bounds for the minimax *expected* regret of the Gaussian prior-based Thompson Sampling algorithm in the $K$-armed problem, see [2].

was made in [32, 12], however, deterministic algorithms like UCB remain outside the ambit of such analysis, on account of discontinuities. Theorem 4 provides a complete characterization of this limit.

**Theorem 4 (Diffusion asymptotics for canonical UCB)** *Suppose that the mean reward of arm $i \in \{1, 2\}$ is given by $\mu_i = \mu + \theta_i/\sqrt{n}$, where $n$ is the horizon of play and $\mu, \theta_1, \theta_2 \geqslant 0$ are fixed constants, and reward variances are $\sigma_1^2, \sigma_2^2$. Define $\Delta_0 := |\theta_1 - \theta_2|$. Denote the cumulative reward earned from arm $i$ until time $m$ by $S_{i,m} := \sum_{j=1}^{N_i(m)} X_{i,j}$, and let $\tilde{S}_{i,m} := S_{i,m} - \mu N_i(m)$. Then, the following process-level convergences hold under the policy $\pi$ given by Algorithm 1 with $\rho > 1$:*

$$(I) \quad \left( \frac{\tilde{S}_{1,\lfloor nt \rfloor}}{\sqrt{n}}, \frac{\tilde{S}_{2,\lfloor nt \rfloor}}{\sqrt{n}} \right) \Rightarrow \left( \frac{\theta_1 t}{2} + \frac{\sigma_1}{\sqrt{2}} B_1(t), \frac{\theta_2 t}{2} + \frac{\sigma_2}{\sqrt{2}} B_2(t) \right),$$

$$(II) \quad \frac{R^\pi_{\lfloor nt \rfloor}}{\sqrt{n}} \Rightarrow \frac{\Delta_0 t}{2} + \sqrt{\frac{\sigma_1^2 + \sigma_2^2}{2}} \tilde{B}(t),$$

*where the process-level convergence is over $t \in [0, 1]$, and $B_1(t)$ and $B_2(t)$ are independent standard Brownian motions in $\mathbb{R}$, and $\tilde{B}(t) := -\sqrt{\frac{\sigma_1^2}{\sigma_1^2 + \sigma_2^2}} B_1(t) - \sqrt{\frac{\sigma_2^2}{\sigma_1^2 + \sigma_2^2}} B_2(t)$.*

**Proof sketch.** Note that if the arms are played $\lfloor n/2 \rfloor$ times each *independently* over the horizon of play $n$ (resulting in $N_i(n) = \lfloor n/2 \rfloor$, $i \in \{1, 2\}$), part (I) of the stated assertion would immediately follow from Donsker's Theorem (see [8], Section 14). However, since the sequence of plays, and hence also the eventual allocation $(N_1(n), N_2(n))$, is determined *adaptively* by the policy, the aforementioned convergence may no longer be true. Here, the result hinges crucially on the observation from Theorem 1 that for any arm $i \in \{1, 2\}$ as $n \to \infty$, $N_i(n)/n \xrightarrow{p} 1/2$ under UCB when $\Delta \asymp 1/\sqrt{n}$ (diffusion-scaled gaps are covered under the "small gap" regime). This observation facilitates a standard "random time-change" argument $t \leftarrow N_i(\lfloor nt \rfloor)/n$, $i \in \{1, 2\}$, which followed upon by an application of Donsker's Theorem, leads to the stated assertion in (I). This has the profound implication that for diffusion-scaled gaps, a two-armed bandit under UCB is, in fact, well-approximated by a classical system with *independent* samples (sample-interdependence due to the adaptive nature of the policy is washed away in the limit). The conclusion in (II) follows after a direct application of the Continuous Mapping Theorem (see [8], Theorem 2.7) to (I). □

**Discussion.** An immediate observation following Theorem 4 is that the normalized regret $R^\pi_n/\sqrt{n}$ is asymptotically Normal with mean $\Delta_0/2$ and variance $\left(\sigma_1^2 + \sigma_2^2\right)/2$ under UCB. Apart from aiding in obvious inferential tasks like construction of (asymptotically valid) confidence intervals (see, e.g., the binary hypothesis testing example referenced in Figure 2(c)), etc., such information provides new insights as to the problem's minimax complexity as well. This is because $\Delta \asymp 1/\sqrt{n}$ is known to be the information-theoretic "worst-case" for the problem; the smallest achievable regret in this regime must asymptotically be dominated by that under UCB, i.e., $\Delta_0 \sqrt{n}/2$. It is also noteworthy that while the diffusion limit in Theorem 4 does not itself depend on the exploration coefficient $\rho$, the rate at which the system converges to said limit indeed depends on $\rho$. Theorem 4 will continue to hold only as long as $\rho = \omega\left((\log \log n)/\log n\right)$; for smaller $\rho$, the convergence of $N_i(n)/n$ to $1/2$ may no longer be true (refer to the proof of Theorem 1 in the "small" gap regime in Appendix E).

## 4 Concluding remarks and open problems

Our results for the two-armed problem under canonical UCB (Algorithm 1) may be generalizable to the $K$-armed setting, leveraging the observations and insights from Theorem 1 (a simple extension is provided in Appendix B). The $K$-armed problem under UCB is of interest in its own right: we postulate a division of sampling effort within and across clusters of "similar" and "separated" arms, determined by their relative sub-optimality gaps in the spirit of Theorem 1. We expect that similar generalizations are possible also for Theorem 3 and Theorem 4. For Thompson Sampling, on the other hand, things are less obvious even in the two-armed setting. For example, in spite of compelling numerical evidence (refer, e.g., to Figure 1(b)) suggesting a plausibly non-degenerate distribution of arm-sampling rates for bounded rewards in $[0, 1]$ with means away from 0, the proof of Theorem 2 relies heavily on the rewards being deterministic $0/1$, and cannot be extended to the general case. In addition, similar results are conjectured also for the more widely used Gaussian prior-based version of the algorithm. Such results may, in future, shed light on several important diffusion-limit performance metrics of Thompson Sampling, and elucidate its normalized minimax (stochastic) regret behavior.

## Acknowledgments and Disclosure of Funding

The authors sincerely thank the anonymous reviewers for their insightful remarks and constructive feedback on the initial submission, and hope that this revision satisfactorily addresses all their points. The authors declare an absence of any competing interests, financial or otherwise.

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
