## Supplementary material

**General organization:**

1. Appendix A provides closed-form expressions for the $\lambda_\rho^*(\theta)$ and $h_\rho(\theta)$ functions that appear in Theorem 1 and Theorem 3.

2. Appendix B contains the statement of Theorem 5, which is a simple extension of Theorem 1 to the $K$-armed setting.

3. Appendix C states three ancillary results that will be used in other proofs.

4. Appendix D provides the proof of Theorem 1 in the "large gap" regime.

7. Appendix G provides the proof of Theorem 2.

8. Appendix H provides the proof of Theorem 3.

9. Appendix I provides the proof of Theorem 4.

10. Appendix J provides the proof of Theorem 5 from Appendix B.

11. Appendix K provides proofs for the ancillary results stated in Appendix C.

**Additional notation.** In the proofs that follow, $\lceil \cdot \rceil$ has been used to denote the "ceiling operator," i.e., $\lceil x \rceil = \inf\{\nu \in \mathbb{N} : \nu \geqslant x\}$ for any $x \in \mathbb{R}$. Similarly, $\lfloor \cdot \rfloor$ denotes the "floor operator," i.e., $\lfloor x \rfloor = \sup\{\nu \in \mathbb{N} : \nu \leqslant x\}$ for any $x \in \mathbb{R}$.

## A  Closed-form expressions for $\lambda_\rho^*(\theta)$ and $h_\rho(\theta)$

$\lambda_\rho^*(\theta)$ is given by:

$$\lambda_\rho^*(\theta) = \frac{1}{2} + \sqrt{\frac{1}{4} - \frac{1}{\left(1 + \sqrt{1 + \frac{\theta}{\rho}}\right)^2}}. \tag{4}$$

$h_\rho(\theta)$ is given by:

$$h_\rho(\theta) = \sqrt{\theta}\left(\frac{1}{2} - \sqrt{\frac{1}{4} - \frac{1}{\left(1 + \sqrt{1 + \frac{\theta}{\rho}}\right)^2}}\right). \tag{5}$$

## B  Arm-sampling rates under UCB in the $K$-armed bandit with multiple optimal arms

---
**Algorithm 3** The canonical UCB policy for $K$-armed bandits.

---
1: **Input:** Exploration coefficient $\rho \in \mathbb{R}_+$.
2: At $t = 1, ..., K$, play each arm $i \in \{1, ..., K\}$ once.
3: **for** $t \in \{K+1, K+2, ...\}$ **do**
4:   Play arm $\pi_t \in \arg\max_{i \in \{1,...,K\}}\left(\bar{X}_i(t-1) + \sqrt{\frac{\rho \log(t-1)}{N_i(t-1)}}\right)$.

---

**Theorem 5 (Asymptotic sampling rate of optimal arms)** *Fix $K \in \mathbb{N}$, and consider a $K$-armed model with arms indexed by $[K] := \{1, ..., K\}$. Let $\mathcal{I} \subseteq [K]$ be the set of optimal arms, i.e., arms with mean $\max_{i \in [K]} \mu_i$. If $\mathcal{I} \neq [K]$, define $\Delta_{\min} := \max_{i \in [K]} \mu_i - \max_{i \in [K] \setminus \mathcal{I}} \mu_i$. Then, there exists a finite $\rho_0 > 1$ that depends only on $|\mathcal{I}|$, such that the following results hold for any arm $i \in \mathcal{I}$ as $n \to \infty$ under Algorithm 3 initialized with $\rho \geqslant \rho_0$:*

*(I) If $\mathcal{I} = [K]$, then*

$$\frac{N_i(n)}{n} \xrightarrow{p} \frac{1}{K}.$$

*(II) If $\mathcal{I} \neq [K]$ and optimal arms are "well-separated," i.e., $\Delta_{\min} = \omega\left(\sqrt{\frac{\log n}{n}}\right)$, then*

$$\frac{N_i(n)}{n} \xrightarrow{p} \frac{1}{|\mathcal{I}|}.$$

**Discussion.** The main observation here is that if the set of optimal arms is "sufficiently separated" from the sub-optimal arms, then classical UCB policies eventually allocate the sampling effort over the set of optimal arms *uniformly*, in probability. This is a desirable property to have from a fairness standpoint, and also markedly different from the instability and imbalance results for Thompson Sampling discussed earlier in Theorem 2. We remark that the $\rho \geqslant \rho_0$ condition is only necessary for tractability of the proof, and conjecture the result to hold, in fact, for any $\rho > 1$, akin to the result for the two-armed setting (Theorem 1). We also conjecture analogous results for "small gap" and "moderate gap" regimes, in the spirit of Theorem 1; proofs, however, can be unwieldy in the general $K$-armed setting. The detailed proof of Theorem 5 is provided in Appendix J.

## C  Auxiliary results

We will frequently use the following version of the Chernoff-Hoeffding inequality [17] in our proofs:

**Fact 1 (Chernoff-Hoeffding bound)** *Suppose that $\{Y_{i,j} : i \in \{1, 2\}, j \in \mathbb{N}\}$ is a collection of independent, zero-mean random variables such that $\forall\, i \in \{1, 2\}, j \in \mathbb{N}$, $Y_{i,j} \in [c_i, 1 + c_i]$ almost surely, for some fixed $c_1, c_2 \leqslant 0$. Then, for any $m_1, m_2 \in \mathbb{N}$ and $\alpha > 0$,*

$$\mathbb{P}\left(\frac{\sum_{j=1}^{m_1} Y_{1,j}}{m_1} - \frac{\sum_{j=1}^{m_2} Y_{2,j}}{m_2} \geqslant \alpha\right) \leqslant \exp\left(\frac{-2\alpha^2 m_1 m_2}{m_1 + m_2}\right).$$

**Proof.** Let $[n] := \{1, ..., n\}$ for $n \in \mathbb{N}$. The Chernoff-Hoeffding inequality in its standard form states that for independent, zero-mean, bounded random variables $\{Z_j : j \in [n]\}$ with $Z_j \in [a_j, b_j] \ \forall\, j \in [n]$, the following holds for any $\varepsilon > 0$,

$$\mathbb{P}\left(\sum_{j=1}^{n} Z_j \geqslant \varepsilon n\right) \leqslant \exp\left(\frac{-2\varepsilon^2 n^2}{\sum_{j=1}^{n} (b_j - a_j)^2}\right). \tag{6}$$

The desired form of the inequality can be obtained by making the following substitutions in (6): $n \leftarrow m_1 + m_2$; $Z_j \leftarrow \frac{Y_{1,j}}{m_1}$, $a_j \leftarrow \frac{c_1}{m_1}$, $b_j \leftarrow \frac{1+c_1}{m_1}$ for $j \in [m_1]$; $Z_j \leftarrow \frac{-Y_{2,j-m_1}}{m_2}$, $a_j \leftarrow \frac{-(1+c_2)}{m_2}$, $b_j \leftarrow \frac{-c_2}{m_2}$ for $j \in [m_1 + m_2] \setminus [m_1]$; and $\varepsilon \leftarrow \frac{\alpha}{m_1 + m_2}$, in that order. $\square$

In addition, we will use in the proof of Theorem 2 the following two properties of the Beta distribution:

**Fact 2** *If $\theta_k, \tilde{\theta}_k$ are $Beta(1, k+1)$-distributed, and $\theta_k, \tilde{\theta}_l$ are independent $\forall\, k, l \in \mathbb{N} \cup \{0\}$, then*

$$\mathbb{P}\left(\theta_k > \tilde{\theta}_l\right) = \frac{l+1}{k+l+2} \quad \text{for any } k, l \in \mathbb{N} \cup \{0\}.$$

**Fact 3** *If $\theta_k, \tilde{\theta}_k$ are $Beta(k+1, 1)$-distributed, and $\theta_k, \tilde{\theta}_l$ are independent $\forall\, k, l \in \mathbb{N} \cup \{0\}$, then*

$$\mathbb{P}\left(\theta_k > \tilde{\theta}_l\right) = \frac{k+1}{k+l+2} \quad \text{for any } k, l \in \mathbb{N} \cup \{0\}.$$

The proofs for Fact (2) and Fact (3) are elementary, and provided in Appendix K.

# D    Proof of Theorem 1 in the "large gap" regime

The proof is straightforward in this regime. We know that for $\rho > 1$, $\mathbb{E} R_n^\pi \leqslant C\rho \left( \frac{\log n}{\Delta} + \frac{\Delta}{\rho - 1} \right)$ for some absolute constant $C$ (see [6], Theorem 7). Since $\mathbb{E} R_n^\pi = \Delta \mathbb{E}\left[n - N_{i^*}(n)\right]$, it follows that $\mathbb{E}\left[\frac{n - N_{i^*}(n)}{n}\right] = o(1)$ in the "large gap" regime. Using Markov's inequality, we then conclude that $\frac{n - N_{i^*}(n)}{n} = o_p(1)$, or equivalently, $\lim_{n \to \infty} \frac{N_{i^*}(n)}{n} = 1$. Results for "small" and "moderate" gaps are provided separately in Appendix E and Appendix F respectively.    □

# E    Proof of Theorem 1 in the "small gap" regime

Without loss of generality, suppose that arm 1 is optimal, i.e., $\mu_1 \geqslant \mu_2$. We will show that for any $\epsilon > 0$, it follows that $\lim_{n \to \infty} \mathbb{P}\left(\frac{N_1(n)}{n} \geqslant \frac{1}{2} + \epsilon\right) = 0$. Then, since arm 2 is inferior, an identical result would naturally hold for it as well. Combining the two would prove our assertion as desired. To this end, pick an arbitrary $\epsilon \in (0, 1/2)$, define $u(n) := \left\lceil \left(\frac{1}{2} + \epsilon\right) n \right\rceil$, and consider the following:

$$
\begin{aligned}
N_1(n) &\leqslant u(n) + \sum_{t=u(n)+1}^{n} \mathbb{1}\left\{\pi_t = 1,\ N_1(t-1) \geqslant u(n)\right\} \qquad\qquad \text{(this is always true)} \\
&= u(n) + \sum_{t=u(n)}^{n-1} \mathbb{1}\left\{\pi_{t+1} = 1,\ N_1(t) \geqslant u(n)\right\} \\
&\leqslant u(n) + \sum_{t=u(n)}^{n-1} \mathbb{1}\left\{\pi_{t+1} = 1,\ N_1(t) \geqslant u(t)\right\} \\
&\leqslant u(n) + \sum_{t=u(n)}^{n-1} \mathbb{1}\left\{\bar{X}_1(t) - \bar{X}_2(t) \geqslant \sqrt{\rho \log t}\left(\frac{1}{\sqrt{N_2(t)}} - \frac{1}{\sqrt{N_1(t)}}\right),\ N_1(t) \geqslant u(t)\right\} \\
&= u(n) + \underbrace{\sum_{t=u(n)}^{n-1} \mathbb{1}\left\{\bar{Y}_1(t) - \bar{Y}_2(t) \geqslant \sqrt{\rho \log t}\left(\frac{1}{\sqrt{N_2(t)}} - \frac{1}{\sqrt{N_1(t)}}\right) - \Delta,\ N_1(t) \geqslant u(t)\right\}}_{=:Z(n)},
\end{aligned}
$$
(7)

where $\bar{Y}_i(t) := \frac{\sum_{j=1}^{N_i(t)} Y_{i,j}}{N_i(t)}$ with $Y_{i,j} := X_{i,j} - \mu_i$, $i \in \{1, 2\}, j \in \mathbb{N}$. Clearly, $Y_{i,j}$'s are independent, zero-mean, and $Y_{i,j} \in [-\mu_i, 1 - \mu_i] \ \forall\, i \in \{1, 2\}, j \in \mathbb{N}$.

## E.1    An almost sure lower bound on the arm-sampling rates

As a meta-result, we will first show that $N_i(n)/n$, for both arms $i \in \{1, 2\}$, is bounded away from $0$ by a positive constant, almost surely. To this end, consider $n$ large enough such that for the $\epsilon$ selected earlier, we have $\Delta < \sqrt{\frac{\rho \log n}{n}} \left(\frac{1}{\sqrt{1/2 - \epsilon}} - \frac{1}{\sqrt{1/2 + \epsilon}}\right)$; this is possible since $\Delta = o\left(\sqrt{\frac{\log n}{n}}\right)$ in the "small gap" regime. Working with a large enough $n$ will allow us to use the Chernoff-Hoeffding bound (Fact 1) in step $(\star)$ in the forthcoming analysis. Observe from (7) that

$$\mathbb{E}Z(n)$$

$$= \sum_{t=u(n)}^{n-1} \mathbb{P}\left(\bar{Y}_1(t) - \bar{Y}_2(t) \geqslant \sqrt{\rho \log t}\left(\frac{1}{\sqrt{N_2(t)}} - \frac{1}{\sqrt{N_1(t)}}\right) - \Delta, \ N_1(t) \geqslant u(t)\right)$$

$$= \sum_{t=u(n)}^{n-1} \sum_{m=u(t)}^{t-1} \mathbb{P}\left(\frac{\sum_{j=1}^m Y_{1,j}}{m} - \frac{\sum_{j=1}^{t-m} Y_{2,j}}{t-m} \geqslant \sqrt{\rho \log t}\left(\frac{1}{\sqrt{t-m}} - \frac{1}{\sqrt{m}}\right) - \Delta, \ N_1(t) = m\right)$$

$$\underset{(\star)}{\leqslant} \sum_{t=u(n)}^{n-1} \sum_{m=u(t)}^{t-1} \mathbb{P}\left(\frac{\sum_{j=1}^m Y_{1,j}}{m} - \frac{\sum_{j=1}^{t-m} Y_{2,j}}{t-m} \geqslant \sqrt{\rho \log t}\left(\frac{1}{\sqrt{t-m}} - \frac{1}{\sqrt{m}}\right) - \Delta\right)$$

$$\underset{(\dagger)}{\leqslant} \sum_{t=u(n)}^{n-1} \sum_{m=u(t)}^{t-1} \exp\left[-2\rho\left(1 - 2\sqrt{\frac{m}{t}\left(1 - \frac{m}{t}\right)}\right)\log t\right] \exp\left[4\Delta\sqrt{\rho t \log t}\left(\sqrt{\frac{m}{t}} - \sqrt{1 - \frac{m}{t}}\right)\sqrt{\frac{m}{t}\left(1 - \frac{m}{t}\right)}\right]$$

$$\underset{(\ddagger)}{\leqslant} \sum_{t=u(n)}^{n-1} \sum_{m=u(t)}^{t-1} \exp\left[-2\rho\left(1 - \sqrt{1 - 4\epsilon^2}\right)\log t\right] \exp\left[4\Delta\sqrt{\rho t \log t}\left(\sqrt{\frac{m}{t}} - \sqrt{1 - \frac{m}{t}}\right)\sqrt{\frac{m}{t}\left(1 - \frac{m}{t}\right)}\right]$$

$$\leqslant \sum_{t=u(n)}^{n-1} \sum_{m=u(t)}^{t-1} \exp\left[-2\rho\left(1 - \sqrt{1 - 4\epsilon^2}\right)\log t\right] \exp\left[4\Delta\sqrt{\rho t \log t}\right]$$

$$\leqslant \sum_{t=u(n)}^{n-1} \sum_{m=u(t)}^{t-1} \exp\left[-2\rho\left(1 - \sqrt{1 - 4\epsilon^2}\right)\log t\right] \exp\left[4\Delta\sqrt{\rho n \log n}\right]$$

$$\leqslant \exp\left[4\Delta\sqrt{\rho n \log n}\right] \sum_{t=u(n)}^{n-1} t^{-\left(2\rho - 1 - 2\rho\sqrt{1 - 4\epsilon^2}\right)}$$

$$= \exp\left[o\left(4\sqrt{\rho}\log n\right)\right] \sum_{t=u(n)}^{n-1} t^{-\left(2\rho - 1 - 2\rho\sqrt{1 - 4\epsilon^2}\right)} \qquad \left(\because \Delta = o\left(\sqrt{\frac{\log n}{n}}\right)\right)$$

$$\leqslant n^{\frac{1}{2} - \epsilon} \sum_{t=u(n)}^{n-1} t^{-\left(2\rho - 1 - 2\rho\sqrt{1 - 4\epsilon^2}\right)}, \tag{8}$$

where ($\dagger$) follows after an application of the Chernoff-Hoeffding bound (Fact 1), ($\ddagger$) since $\frac{m}{t}\left(1 - \frac{m}{t}\right) \leqslant \frac{1}{4} - \epsilon^2$ on the interval $\{m : u(t) \leqslant m \leqslant t - 1\}$, and the last inequality in (8) holds for $n$ large enough. Now consider an arbitrary $\delta > 0$. Then,

$$\mathbb{P}\left(N_1(n) - u(n) \geqslant \delta n\right) \leqslant \mathbb{P}\left(Z(n) \geqslant \delta n\right) \qquad \text{(using (7))}$$

$$\leqslant \frac{\mathbb{E}Z(n)}{\delta n} \qquad \text{(Markov's inequality)}$$

$$\leqslant \left(\frac{n^{-\left(\frac{1}{2} + \epsilon\right)}}{\delta}\right) \sum_{t=u(n)}^{n-1} t^{-\left(2\rho - 1 - 2\rho\sqrt{1 - 4\epsilon^2}\right)} \qquad \text{(using (8))}$$

$$\implies \mathbb{P}\left(\frac{N_1(n)}{n} \geqslant \frac{1}{2} + \epsilon + \delta + \frac{1}{n}\right) \leqslant \left(\frac{n^{-\left(\frac{1}{2} + \epsilon\right)}}{\delta}\right) \sum_{t=\lceil \frac{n}{2} \rceil}^{n-1} t^{-\left(2\rho - 1 - 2\rho\sqrt{1 - 4\epsilon^2}\right)}. \tag{9}$$

Define $g(\rho, \epsilon) := \frac{1}{2} + \epsilon + 2\rho - 1 - 2\rho\sqrt{1 - 4\epsilon^2}$. Since $\rho > 1$ is fixed, and $\epsilon \in (0, 1/2)$ is arbitrary, it is possible to push $\epsilon$ close to $1/2$ to ensure that $g(\rho, \epsilon) > 2$. Therefore, $\exists\, \epsilon_\rho \in (0, 1/2)$ s.t. $g(\rho, \epsilon) > 2$ for $\epsilon \geqslant \epsilon_\rho$. Plugging in $\epsilon = \epsilon_\rho$ in (9), we obtain

$$\mathbb{P}\left(\frac{N_1(n)}{n} \geqslant \frac{1}{2} + \epsilon_\rho + \delta + \frac{1}{n}\right) \leqslant \left(\frac{2^{2\rho - 1}}{\delta}\right) n^{-(g(\rho, \epsilon_\rho) - 1)}.$$

Note that since $\epsilon_\rho < 1/2$, $\exists\, \epsilon'_\rho < 1/2$ s.t. $\epsilon_\rho + 1/n < \epsilon'_\rho$ for $n$ large enough, i.e., the following holds for all $n$ large enough:

$$\mathbb{P}\left(\frac{N_1(n)}{n} \geqslant \frac{1}{2} + \epsilon'_\rho + \delta\right) \leqslant \left(\frac{2^{2\rho-1}}{\delta}\right) n^{-(g(\rho,\epsilon_\rho)-1)}.$$

Finally, since $\delta > 0$ is arbitrary, and $g(\rho, \epsilon_\rho) > 2$, it follows from the Borel-Cantelli Lemma that

$$\limsup_{n\to\infty} \frac{N_1(n)}{n} \leqslant \frac{1}{2} + \epsilon'_\rho < 1 \quad \text{w.p. 1}.$$

By assumption, arm 2 is inferior; the above result thus holds, in fact, for both the arms (An almost identical proof can be replicated for rigor). Therefore, we conclude

$$\liminf_{n\to\infty} \frac{N_i(n)}{n} \geqslant \frac{1}{2} - \epsilon'_\rho > 0 \quad \text{w.p. 1} \quad \forall\, i \in \{1, 2\}. \tag{10}$$

## E.2 Closing the loop

In this part of the proof, we will leverage (10) to finally show that $N_i(n)/n = 1/2 + o_p(1)$ for $i \in \{1, 2\}$. To this end, recall from (7) that

$$\mathbb{E}Z(n)$$

$$= \sum_{t=u(n)}^{n-1} \mathbb{P}\left(\bar{Y}_1(t) - \bar{Y}_2(t) \geqslant \sqrt{\rho \log t}\left(\frac{1}{\sqrt{N_2(t)}} - \frac{1}{\sqrt{N_1(t)}}\right) - \Delta,\ N_1(t) \geqslant u(t)\right)$$

$$\leqslant \sum_{t=u(n)}^{n-1} \mathbb{P}\left(\bar{Y}_1(t) - \bar{Y}_2(t) \geqslant \sqrt{\frac{\rho \log t}{t}}\left(\frac{1}{\sqrt{\frac{1}{2}-\epsilon}} - \frac{1}{\sqrt{\frac{1}{2}+\epsilon}}\right) - \Delta,\ N_1(t) \geqslant u(t)\right)$$

$$\leqslant \sum_{t=u(n)}^{n-1} \mathbb{P}\left(\bar{Y}_1(t) - \bar{Y}_2(t) \geqslant \sqrt{\frac{\rho \log t}{t}}\left(\frac{1}{\sqrt{\frac{1}{2}-\epsilon}} - \frac{1}{\sqrt{\frac{1}{2}+\epsilon}}\right) - \Delta\right)$$

$$= \sum_{t=u(n)}^{n-1} \mathbb{P}\left(\underbrace{\sqrt{\frac{t}{\rho \log t}}\left(\bar{Y}_1(t) - \bar{Y}_2(t)\right)}_{=:W_t} \geqslant \sqrt{\frac{2}{1-2\epsilon}} - \sqrt{\frac{2}{1+2\epsilon}} - \Delta\sqrt{\frac{t}{\rho \log t}}\right)$$

$$\leqslant \sum_{t=u(n)}^{n-1} \mathbb{P}\left(W_t \geqslant \frac{1}{\sqrt{1-2\epsilon}} - \frac{1}{\sqrt{1+2\epsilon}}\right), \tag{11}$$

for $n$ large enough; the last inequality following since $\Delta = o\left(\sqrt{\frac{\log n}{n}}\right)$ and $u(n) > n/2$. Now,

$$|W_t|$$

$$\leqslant \sqrt{\frac{t}{\rho \log t}}\left(\left|\frac{\sum_{j=1}^{N_1(t)} Y_{1,j}}{N_1(t)}\right| + \left|\frac{\sum_{j=1}^{N_2(t)} Y_{2,j}}{N_2(t)}\right|\right)$$

$$= \sqrt{\frac{2t}{\rho \log t}}\left(\sqrt{\frac{\log\log N_1(t)}{N_1(t)}}\left|\frac{\sum_{j=1}^{N_1(t)} Y_{1,j}}{\sqrt{2N_1(t)\log\log N_1(t)}}\right| + \sqrt{\frac{\log\log N_2(t)}{N_2(t)}}\left|\frac{\sum_{j=1}^{N_2(t)} Y_{2,j}}{\sqrt{2N_2(t)\log\log N_2(t)}}\right|\right)$$

$$\leqslant \sqrt{\frac{2t}{\rho \log t}}\left(\sqrt{\frac{\log\log t}{N_1(t)}}\left|\frac{\sum_{j=1}^{N_1(t)} Y_{1,j}}{\sqrt{2N_1(t)\log\log N_1(t)}}\right| + \sqrt{\frac{\log\log t}{N_2(t)}}\left|\frac{\sum_{j=1}^{N_2(t)} Y_{2,j}}{\sqrt{2N_2(t)\log\log N_2(t)}}\right|\right)$$

$$= \sqrt{\frac{2\log\log t}{\rho \log t}}\left(\sqrt{\frac{t}{N_1(t)}}\left|\frac{\sum_{j=1}^{N_1(t)} Y_{1,j}}{\sqrt{2N_1(t)\log\log N_1(t)}}\right| + \sqrt{\frac{t}{N_2(t)}}\left|\frac{\sum_{j=1}^{N_2(t)} Y_{2,j}}{\sqrt{2N_2(t)\log\log N_2(t)}}\right|\right). \tag{12}$$

We know that $N_i(t)$, for both arms $i \in \{1, 2\}$, can be lower bounded *path-wise* by a deterministic monotone increasing function of $t$, say $f(t)$, that grows to $+\infty$ as $t \to \infty$. This is a trivial consequence of the structure of canonical UCB (Algorithm 1), and the fact that the rewards are uniformly bounded. We therefore have for any $i \in \{1, 2\}$ that

$$\left| \frac{\sum_{j=1}^{N_i(t)} Y_{i,j}}{\sqrt{2 N_i(t) \log \log N_i(t)}} \right| \leqslant \sup_{m \geqslant f(t)} \left| \frac{\sum_{j=1}^{m} Y_{i,j}}{\sqrt{2m \log \log m}} \right|.$$

For a fixed arm $i \in \{1, 2\}$, $\{Y_{i,j} : j \in \mathbb{N}\}$ is a collection of i.i.d. random variables with $\mathbb{E} Y_{i,1} = 0$ and $\text{Var}(Y_{i,1}) = \text{Var}(X_{i,1}) \leqslant 1$. Also, $f(t)$ is monotone increasing and coercive in $t$. Therefore, the *Law of the Iterated Logarithm* (see [11], Theorem 8.5.2) implies

$$\limsup_{t \to \infty} \left| \frac{\sum_{j=1}^{N_i(t)} Y_{i,j}}{\sqrt{2 N_i(t) \log \log N_i(t)}} \right| \leqslant 1 \quad \text{w.p. } 1 \ \forall \, i \in \{1, 2\}. \tag{13}$$

Using (10), (12) and (13), we conclude that

$$\lim_{t \to \infty} W_t = 0 \quad \text{w.p. } 1. \tag{14}$$

Now consider an arbitrary $\delta > 0$. Then,

$$\mathbb{P}\left(N_1(n) - u(n) \geqslant \delta n\right) \leqslant \mathbb{P}\left(Z(n) \geqslant \delta n\right) \qquad \text{(using (7))}$$

$$\leqslant \frac{\mathbb{E} Z(n)}{\delta n} \qquad \text{(Markov's inequality)}$$

$$\leqslant \frac{1}{\delta n} \sum_{t=u(n)}^{n-1} \mathbb{P}\left(W_t \geqslant \frac{1}{\sqrt{1-2\epsilon}} - \frac{1}{\sqrt{1+2\epsilon}}\right). \qquad \text{(using (11))}$$

$$\leqslant \frac{1}{\delta} \sup_{t > n/2} \mathbb{P}\left(W_t \geqslant \frac{1}{\sqrt{1-2\epsilon}} - \frac{1}{\sqrt{1+2\epsilon}}\right)$$

$$\implies \mathbb{P}\left(\frac{N_1(n)}{n} \geqslant \frac{1}{2} + \epsilon + \delta + \frac{1}{n}\right) \leqslant \frac{1}{\delta} \sup_{t > n/2} \mathbb{P}\left(W_t \geqslant \frac{1}{\sqrt{1-2\epsilon}} - \frac{1}{\sqrt{1+2\epsilon}}\right).$$

Since $\epsilon, \delta > 0$ are arbitrary, it follows that for $n$ large enough,

$$\mathbb{P}\left(\frac{N_1(n)}{n} \geqslant \frac{1}{2} + 2(\epsilon + \delta)\right) \leqslant \frac{1}{\delta} \sup_{t > n/2} \mathbb{P}\left(W_t \geqslant \frac{1}{\sqrt{1-2\epsilon}} - \frac{1}{\sqrt{1+2\epsilon}}\right). \tag{15}$$

Using (14) and (15), we conclude that for any arbitrary $\epsilon, \delta > 0$,

$$\limsup_{n \to \infty} \mathbb{P}\left(\frac{N_1(n)}{n} \geqslant \frac{1}{2} + 2(\epsilon + \delta)\right) \leqslant \frac{1}{\delta} \limsup_{n \to \infty} \mathbb{P}\left(W_n \geqslant \frac{1}{\sqrt{1-2\epsilon}} - \frac{1}{\sqrt{1+2\epsilon}}\right) = 0.$$

It therefore follows that for any $\epsilon' > 0$, $\lim_{n \to \infty} \mathbb{P}\left(\frac{N_1(n)}{n} \geqslant \frac{1}{2} + \epsilon'\right) = 0$; equivalently, $\lim_{n \to \infty} \mathbb{P}\left(\frac{N_2(n)}{n} \leqslant \frac{1}{2} - \epsilon'\right) = 0$. Since arm 2 is inferior by assumption, it naturally holds that $\lim_{n \to \infty} \mathbb{P}\left(\frac{N_2(n)}{n} \geqslant \frac{1}{2} + \epsilon'\right) = 0$ (The steps in E.2 can be replicated near-identically for rigor). Thus, the stated assertion $\frac{N_i(n)}{n} \xrightarrow[n \to \infty]{p} \frac{1}{2} \ \forall \, i \in \{1, 2\}$, follows. $\qquad \square$

## F  Proof of Theorem 1 in the "moderate gap" regime

Firstly, note that the $\lambda_\rho^*(\theta)$ that solves (2), satisfies the following properties: (i) Continuous and monotone increasing in $\theta \geqslant 0$, (ii) $\lambda_\rho^*(\theta) \geqslant 1/2$ for all $\theta \geqslant 0$, (iii) $\lambda_\rho^*(0) = 1/2$ and $\lambda_\rho^*(\theta) \to 1$ as $\theta \to \infty$.

Secondly, because we are only interested in asymptotics, the $\Delta \sim \sqrt{\frac{\theta \log n}{n}}$ condition is as good as $\Delta = \sqrt{\frac{\theta \log n}{n}}$, since for any arbitrarily small $\epsilon' > 0$, $\Delta \in \left( \sqrt{\frac{(\theta - \epsilon') \log n}{n}}, \sqrt{\frac{(\theta + \epsilon') \log n}{n}} \right)$ for $n$ large enough; the stated assertion would follow in the limit as $\epsilon'$ approaches $0$. In what follows, we will therefore assume for readability of the proof, and without loss of generality, that $\Delta = \sqrt{\frac{\theta \log n}{n}}$.

Thirdly, without loss of generality, suppose that arm 1 is optimal, i.e., $\mu_1 \geqslant \mu_2$.

### F.1 Focusing on arm 1

Consider an arbitrary $\epsilon \in \left( 0, 1 - \lambda_\rho^*(\theta) \right)$, and define $u(n) := \left\lceil \left( \lambda_\rho^*(\theta) + \epsilon \right) n \right\rceil$. We know that

$$N_1(n) \leqslant u(n) + \sum_{t=u(n)+1}^{n} \mathbb{1}\left\{ \pi_t = 1, \ N_1(t-1) \geqslant u(n) \right\} \qquad \text{(this is always true)}$$

$$= u(n) + \sum_{t=u(n)}^{n-1} \mathbb{1}\left\{ \pi_{t+1} = 1, \ N_1(t) \geqslant u(n) \right\}$$

$$\leqslant u(n) + \sum_{t=u(n)}^{n-1} \mathbb{1}\left\{ \pi_{t+1} = 1, \ N_1(t) \geqslant u(t) \right\}$$

$$\leqslant u(n) + \sum_{t=u(n)}^{n-1} \mathbb{1}\left\{ \bar{X}_1(t) - \bar{X}_2(t) \geqslant \sqrt{\rho \log t} \left( \frac{1}{\sqrt{N_2(t)}} - \frac{1}{\sqrt{N_1(t)}} \right), \ N_1(t) \geqslant u(t) \right\}$$

$$= u(n) + \underbrace{\sum_{t=u(n)}^{n-1} \mathbb{1}\left\{ \bar{Y}_1(t) - \bar{Y}_2(t) \geqslant \sqrt{\rho \log t} \left( \frac{1}{\sqrt{N_2(t)}} - \frac{1}{\sqrt{N_1(t)}} \right) - \Delta, \ N_1(t) \geqslant u(t) \right\}}_{=:Z(n)},$$

$$\tag{16}$$

where $\bar{Y}_i(t) := \frac{\sum_{j=1}^{N_i(t)} Y_{i,j}}{N_i(t)}$ with $Y_{i,j} := X_{i,j} - \mu_i$, $i \in \{1,2\}$, $j \in \mathbb{N}$. Clearly, $Y_{i,j}$'s are independent, zero-mean, and $Y_{i,j} \in [-\mu_i, 1 - \mu_i] \ \forall \, i \in \{1,2\}, j \in \mathbb{N}$.

#### F.1.1 An almost sure lower bound on the arm-sampling rates

Consider $n$ large enough such that $\sqrt{\frac{\log n}{n}}$ is monotone decreasing in $n$ ($n \geqslant 3$ suffices). This will enable the inequality in step (†) below. From (16), we have

$$\mathbb{E} Z(n)$$

$$= \sum_{t=u(n)}^{n-1} \mathbb{P}\left( \bar{Y}_1(t) - \bar{Y}_2(t) \geqslant \sqrt{\rho \log t} \left( \frac{1}{\sqrt{N_2(t)}} - \frac{1}{\sqrt{N_1(t)}} \right) - \Delta, \ N_1(t) \geqslant u(t) \right)$$

$$= \sum_{t=u(n)}^{n-1} \mathbb{P}\left( \bar{Y}_1(t) - \bar{Y}_2(t) \geqslant \sqrt{\rho \log t} \left( \frac{1}{\sqrt{N_2(t)}} - \frac{1}{\sqrt{N_1(t)}} \right) - \sqrt{\frac{\theta \log n}{n}}, \ N_1(t) \geqslant u(t) \right)$$

$$\underset{(\dagger)}{\leqslant} \sum_{t=u(n)}^{n-1} \mathbb{P}\left( \bar{Y}_1(t) - \bar{Y}_2(t) \geqslant \sqrt{\rho \log t} \left( \frac{1}{\sqrt{N_2(t)}} - \frac{1}{\sqrt{N_1(t)}} \right) - \sqrt{\frac{\theta \log t}{t}}, \ N_1(t) \geqslant u(t) \right)$$

$$= \sum_{t=u(n)}^{n-1} \mathbb{P}\left( \bar{Y}_1(t) - \bar{Y}_2(t) \geqslant \sqrt{\rho \log t} \left( \frac{1}{\sqrt{N_2(t)}} - \frac{1}{\sqrt{N_1(t)}} - \sqrt{\frac{\theta}{\rho t}} \right), \ N_1(t) \geqslant u(t) \right) \tag{17}$$

$$\leqslant \sum_{t=u(n)}^{n-1} \sum_{m=u(t)}^{t-1} \mathbb{P}\left( \frac{\sum_{j=1}^{m} Y_{1,j}}{m} - \frac{\sum_{j=1}^{t-m} Y_{2,j}}{t-m} \geqslant \sqrt{\rho \log t} \left( \frac{1}{\sqrt{t-m}} - \frac{1}{\sqrt{m}} - \sqrt{\frac{\theta}{\rho t}} \right) \right). \tag{18}$$

Notice that in the interval $m \in [u(t), t-1]$,

$$\frac{1}{\sqrt{t-m}} - \frac{1}{\sqrt{m}} - \sqrt{\frac{\theta}{2t}} \geqslant \frac{1}{\sqrt{t-u(t)}} - \frac{1}{\sqrt{u(t)}} - \sqrt{\frac{\theta}{\rho t}} \geqslant \frac{1}{\sqrt{t}} \left( \frac{1}{\sqrt{1-\lambda_\rho^*(\theta)-\epsilon}} - \frac{1}{\sqrt{\lambda_\rho^*(\theta)+\epsilon}} - \sqrt{\frac{\theta}{\rho}} \right) > 0,$$

where the final inequality follows since $\lambda_\rho^*(\theta)$ is the solution to (2). We can therefore apply the Chernoff-Hoeffding bound (Fact 1) to (18) to conclude

$$\mathbb{E}Z(n)$$
$$\leqslant \sum_{t=u(n)}^{n-1} \sum_{m=u(t)}^{t-1} \exp\left[ -2\rho \log t \left( \frac{1}{\sqrt{t-m}} - \frac{1}{\sqrt{m}} - \sqrt{\frac{\theta}{\rho t}} \right)^2 \frac{m(t-m)}{t} \right]$$
$$= \sum_{t=u(n)}^{n-1} \sum_{m=u(t)}^{t-1} \exp\left[ -2\rho \log t \left( \sqrt{\frac{m}{t}} - \sqrt{1-\frac{m}{t}} - \sqrt{\frac{\theta}{\rho}} \sqrt{\frac{m}{t}\left(1-\frac{m}{t}\right)} \right)^2 \right]$$
$$= \sum_{t=u(n)}^{n-1} \sum_{m=u(t)}^{t-1} \exp\left[ -2\rho \log t \left( f\left(\frac{m}{t}\right) \right)^2 \right], \tag{19}$$

where the function $f(x) := \sqrt{x} - \sqrt{1-x} - \sqrt{\theta x(1-x)/\rho}$. Notice that $f(x)$ is monotone increasing over the interval $(1/2, 1)$ ($\because \theta, \rho \geqslant 0$). Also, note that $1/2 < \lambda_\rho^*(\theta)+\epsilon < m/t < 1$ in (19). Thus, we have in (19) that $f\left(\frac{m}{t}\right) \geqslant \min_{x \in [\lambda_\rho^*(\theta)+\epsilon, 1)} f(x) = f\left(\lambda_\rho^*(\theta)+\epsilon\right)$. An expression for $f\left(\lambda_\rho^*(\theta)+\epsilon\right)$ is provided in (22) below. Observe that $f\left(\lambda_\rho^*(\theta)+\epsilon\right) > 0$; this follows since $\lambda_\rho^*(\theta)$ is the solution to (2). Using these facts in (19), we conclude

$$\mathbb{E}Z(n) \leqslant \sum_{t=u(n)}^{n-1} \sum_{m=u(t)}^{t-1} \exp\left[ -2\rho \log t \left( \min_{x \in [\lambda_\rho^*(\theta)+\epsilon, 1)} f(x) \right)^2 \right]$$
$$= \sum_{t=u(n)}^{n-1} \sum_{m=u(t)}^{t-1} \exp\left[ -2\rho \log t \left( f\left(\lambda_\rho^*(\theta)+\epsilon\right) \right)^2 \right]$$
$$\leqslant \sum_{t=u(n)}^{n-1} t^{1-2\rho\left(f\left(\lambda_\rho^*(\theta)+\epsilon\right)\right)^2}$$
$$= \sum_{t=\lceil (\lambda_\rho^*(\theta)+\epsilon)n \rceil}^{n-1} t^{1-2\rho\left(f\left(\lambda_\rho^*(\theta)+\epsilon\right)\right)^2}. \tag{20}$$

Now consider an arbitrary $\delta > 0$. We then have

$$\mathbb{P}\left(N_1(n) - u(n) \geqslant \delta n\right) \leqslant \mathbb{P}\left(Z(n) \geqslant \delta n\right) \qquad \text{(using (16))}$$
$$\leqslant \frac{\mathbb{E}Z(n)}{\delta n} \qquad \text{(Markov's inequality)}$$
$$\implies \mathbb{P}\left(N_1(n) \geqslant \lceil (\lambda_\rho^*(\theta)+\epsilon)n \rceil + \delta n\right) \leqslant \frac{1}{\delta n} \sum_{t=\lceil (\lambda_\rho^*(\theta)+\epsilon)n \rceil}^{n-1} t^{1-2\rho\left(f\left(\lambda_\rho^*(\theta)+\epsilon\right)\right)^2}. \qquad \text{(using (20))}$$
$$\tag{21}$$

Note that $f\left(\lambda_\rho^*(\theta)+\epsilon\right)$ is given by

$$f\left(\lambda_\rho^*(\theta)+\epsilon\right) = \sqrt{\lambda_\rho^*(\theta)+\epsilon} - \sqrt{1-\lambda_\rho^*(\theta)-\epsilon} - \sqrt{\frac{\theta}{\rho}}\sqrt{\left(\lambda_\rho^*(\theta)+\epsilon\right)\left(1-\lambda_\rho^*(\theta)-\epsilon\right)}. \tag{22}$$

Setting $\epsilon = 0$ in (22) yields $f\left(\lambda_\rho^*(\theta)\right) = 0$ (follows from (2)), whereas setting $\epsilon = 1 - \lambda_\rho^*(\theta)$ yields $f(1) = 1$. Since $\rho > 1$, and $f\left(\lambda_\rho^*(\theta)+\epsilon\right)$ is continuous and monotone increasing in $\epsilon$,

$\exists\, \epsilon_{\theta,\rho} \in \left(0, 1 - \lambda_\rho^*(\theta)\right)$ s.t. $f\left(\lambda_\rho^*(\theta) + \epsilon\right) > 1/\sqrt{\rho}$ for $\epsilon \geqslant \epsilon_{\theta,\rho}$. Substituting $\epsilon = \epsilon_{\theta,\rho}$ in (21) and using the aforementioned fact, we obtain

$$
\mathbb{P}\left(N_1(n) \geqslant \left\lceil \left(\lambda_\rho^*(\theta) + \epsilon_{\theta,\rho}\right)n \right\rceil + \delta n\right) \leqslant \frac{1}{\delta n} \sum_{t=\left\lceil \left(\lambda_\rho^*(\theta)+\epsilon_{\theta,\rho}\right)n\right\rceil}^{n-1} t^{1-2\rho\left(f\left(\lambda_\rho^*(\theta)+\epsilon_{\theta,\rho}\right)\right)^2}
$$
$$
\leqslant \left(\frac{2^{2\rho-1}}{\delta}\right) n^{-\left(2\rho\left(f\left(\lambda_\rho^*(\theta)+\epsilon_{\theta,\rho}\right)\right)^2 - 1\right)}, \qquad (23)
$$

where the last inequality follows since $1/\sqrt{\rho} < f\left(\lambda_\rho^*(\theta) + \epsilon_{\theta,\rho}\right) < 1$, and $\lambda_\rho^*(\theta) + \epsilon_{\theta,\rho} > 1/2$. Finally since $\delta > 0$ is arbitrary, we conclude from (23) using the Borel-Cantelli Lemma that

$$
\limsup_{n\to\infty} \frac{N_1(n)}{n} \leqslant \lambda_\rho^*(\theta) + \epsilon_{\theta,\rho} < 1 \quad \text{w.p. } 1.
$$

The above result naturally holds for arm 2 as well, since it is inferior by assumption (we resort to the cop-out that a near-identical argument handles its case). Therefore, in conclusion,

$$
\liminf_{n\to\infty} \frac{N_i(n)}{n} \geqslant 1 - \lambda_\rho^*(\theta) - \epsilon_{\theta,\rho} > 0 \quad \text{w.p. } 1 \quad \forall\, i \in \{1, 2\}. \qquad (24)
$$

### F.1.2 Closing the loop

From (17), we know that

$$
\mathbb{E}Z(n)
$$
$$
\leqslant \sum_{t=u(n)}^{n-1} \mathbb{P}\left(\bar{Y}_1(t) - \bar{Y}_2(t) \geqslant \sqrt{\rho \log t}\left(\frac{1}{\sqrt{N_2(t)}} - \frac{1}{\sqrt{N_1(t)}} - \sqrt{\frac{\theta}{\rho t}}\right),\ N_1(t) \geqslant u(t)\right)
$$
$$
\leqslant \sum_{t=u(n)}^{n-1} \mathbb{P}\left(\bar{Y}_1(t) - \bar{Y}_2(t) \geqslant \sqrt{\frac{\rho \log t}{t}}\left(\frac{1}{\sqrt{1 - \lambda_\rho^*(\theta) - \epsilon}} - \frac{1}{\sqrt{\lambda_\rho^*(\theta) + \epsilon}} - \sqrt{\frac{\theta}{\rho}}\right)\right)
$$
$$
= \sum_{t=u(n)}^{n-1} \mathbb{P}\left(\underbrace{\sqrt{\frac{t}{\rho \log t}}\left(\bar{Y}_1(t) - \bar{Y}_2(t)\right)}_{=:W_t} \geqslant \frac{1}{\sqrt{1 - \lambda_\rho^*(\theta) - \epsilon}} - \frac{1}{\sqrt{\lambda_\rho^*(\theta) + \epsilon}} - \sqrt{\frac{\theta}{\rho}}\right), \qquad (25)
$$

where we already know that $\frac{1}{\sqrt{1-\lambda_\rho^*(\theta)-\epsilon}} - \frac{1}{\sqrt{\lambda_\rho^*(\theta)+\epsilon}} - \sqrt{\frac{\theta}{\rho}} > 0$ (since $\lambda_\rho^*(\theta)$ is the solution to (2)). Now,

$$
|W_t|
$$
$$
\leqslant \sqrt{\frac{t}{\rho \log t}}\left(\left|\frac{\sum_{j=1}^{N_1(t)} Y_{1,j}}{N_1(t)}\right| + \left|\frac{\sum_{j=1}^{N_2(t)} Y_{2,j}}{N_2(t)}\right|\right)
$$
$$
= \sqrt{\frac{2t}{\rho \log t}}\left(\sqrt{\frac{\log\log N_1(t)}{N_1(t)}}\left|\frac{\sum_{j=1}^{N_1(t)} Y_{1,j}}{\sqrt{2N_1(t)\log\log N_1(t)}}\right| + \sqrt{\frac{\log\log N_2(t)}{N_2(t)}}\left|\frac{\sum_{j=1}^{N_2(t)} Y_{2,j}}{\sqrt{2N_2(t)\log\log N_2(t)}}\right|\right)
$$
$$
\leqslant \sqrt{\frac{2t}{\rho \log t}}\left(\sqrt{\frac{\log\log t}{N_1(t)}}\left|\frac{\sum_{j=1}^{N_1(t)} Y_{1,j}}{\sqrt{2N_1(t)\log\log N_1(t)}}\right| + \sqrt{\frac{\log\log t}{N_2(t)}}\left|\frac{\sum_{j=1}^{N_2(t)} Y_{2,j}}{\sqrt{2N_2(t)\log\log N_2(t)}}\right|\right)
$$
$$
= \sqrt{\frac{2\log\log t}{\rho \log t}}\left(\sqrt{\frac{t}{N_1(t)}}\left|\frac{\sum_{j=1}^{N_1(t)} Y_{1,j}}{\sqrt{2N_1(t)\log\log N_1(t)}}\right| + \sqrt{\frac{t}{N_2(t)}}\left|\frac{\sum_{j=1}^{N_2(t)} Y_{2,j}}{\sqrt{2N_2(t)\log\log N_2(t)}}\right|\right). \qquad (26)
$$

We know that $N_i(t)$, for both arms $i \in \{1, 2\}$, can be lower bounded *path-wise* by a deterministic monotone increasing function of $t$, say $g(t)$, that grows to $+\infty$ as $t \to \infty$. This is a trivial consequence

of the structure of the canonical UCB policy (Algorithm 1), and the fact that the rewards are uniformly bounded. Therefore, for any arm $i \in \{1, 2\}$, we have

$$\left| \frac{\sum_{j=1}^{N_i(t)} Y_{i,j}}{\sqrt{2N_i(t) \log \log N_i(t)}} \right| \leqslant \sup_{m \geqslant g(t)} \left| \frac{\sum_{j=1}^{m} Y_{i,j}}{\sqrt{2m \log \log m}} \right|.$$

For a fixed $i \in \{1, 2\}$, $\{Y_{i,j} : j \in \mathbb{N}\}$ is a collection of i.i.d. random variables with $\mathbb{E}Y_{i,1} = 0$ and $\mathrm{Var}(Y_{i,1}) = \mathrm{Var}(X_{i,1}) \leqslant 1$. Also, $g(t)$ is a monotone increasing and coercive function of $t$. Therefore, the *Law of the Iterated Logarithm* (see [11], Theorem 8.5.2) implies

$$\limsup_{t \to \infty} \left| \frac{\sum_{j=1}^{N_i(t)} Y_{i,j}}{\sqrt{2N_i(t) \log \log N_i(t)}} \right| \leqslant 1 \quad \text{w.p. 1} \ \forall i \in \{1, 2\}. \tag{27}$$

Using (24), (26) and (27), we conclude that

$$\lim_{t \to \infty} W_t = 0 \quad \text{w.p. 1}. \tag{28}$$

Now consider an arbitrary $\delta > 0$. We have

$$\mathbb{P}(N_1(n) - u(n) \geqslant \delta n) \leqslant \mathbb{P}(Z(n) \geqslant \delta n) \tag{using (16)}$$

$$\leqslant \frac{\mathbb{E}Z(n)}{\delta n} \tag{Markov's inequality}$$

$$\leqslant \frac{1}{\delta n} \sum_{t=u(n)}^{n-1} \mathbb{P}\left( W_t \geqslant \frac{1}{\sqrt{1 - \lambda_\rho^*(\theta) - \epsilon}} - \frac{1}{\sqrt{\lambda_\rho^*(\theta) + \epsilon}} - \sqrt{\frac{\theta}{\rho}} \right) \tag{using (25)}$$

$$\leqslant \frac{1}{\delta} \sup_{t > n/2} \mathbb{P}\left( W_t \geqslant \frac{1}{\sqrt{1 - \lambda_\rho^*(\theta) - \epsilon}} - \frac{1}{\sqrt{\lambda_\rho^*(\theta) + \epsilon}} - \sqrt{\frac{\theta}{\rho}} \right). \tag{29}$$

Using (28) and (29), it follows that

$$\limsup_{n \to \infty} \mathbb{P}(N_1(n) - u(n) \geqslant \delta n) \leqslant \frac{1}{\delta} \limsup_{n \to \infty} \mathbb{P}\left( W_n \geqslant \frac{1}{\sqrt{1 - \lambda_\rho^*(\theta) - \epsilon}} - \frac{1}{\sqrt{\lambda_\rho^*(\theta) + \epsilon}} - \sqrt{\frac{\theta}{\rho}} \right) = 0.$$

Since $u(n) = \lceil (\lambda_\rho^*(\theta) + \epsilon) n \rceil$ and $\epsilon, \delta > 0$ are arbitrary, we conclude that for any $\epsilon > 0$, it holds that $\lim_{n \to \infty} \mathbb{P}\left( \frac{N_1(n)}{n} \geqslant \lambda_\rho^*(\theta) + \epsilon \right) = 0$. Equivalently, $\lim_{n \to \infty} \mathbb{P}\left( \frac{N_2(n)}{n} \leqslant 1 - \lambda_\rho^*(\theta) - \epsilon \right) = 0$ holds for any $\epsilon > 0$. □

## F.2 Focusing on arm 2 and concluding

We will essentially replicate here the proof for arm 1 given in F.1, albeit with a few subtle modifications to account for the fact that arm 2 is inferior. Consistent with previous approach and notation, we consider an arbitrary $\epsilon \in (0, \lambda_\rho^*(\theta))$ and set $u(n) := \lceil (1 - \lambda_\rho^*(\theta) + \epsilon) n \rceil$, where $\lambda_\rho^*(\theta)$ is the solution to (2) (Note that the definition of $u(n)$ here is different from the one used in the proof for

arm 1.). We know that

$$N_2(n) \leqslant u(n) + \sum_{t=u(n)+1}^{n} \mathbb{1}\left\{\pi_t = 2, \ N_2(t-1) \geqslant u(n)\right\} \qquad \text{(this is always true)}$$

$$= u(n) + \sum_{t=u(n)}^{n-1} \mathbb{1}\left\{\pi_{t+1} = 2, \ N_2(t) \geqslant u(n)\right\}$$

$$\leqslant u(n) + \sum_{t=u(n)}^{n-1} \mathbb{1}\left\{\bar{X}_2(t) - \bar{X}_1(t) \geqslant \sqrt{\rho \log t}\left(\frac{1}{\sqrt{N_1(t)}} - \frac{1}{\sqrt{N_2(t)}}\right), \ N_2(t) \geqslant u(n)\right\}$$

$$= u(n) + \underbrace{\sum_{t=u(n)}^{n-1} \mathbb{1}\left\{\bar{Y}_2(t) - \bar{Y}_1(t) \geqslant \sqrt{\rho \log t}\left(\frac{1}{\sqrt{N_1(t)}} - \frac{1}{\sqrt{N_2(t)}}\right) + \Delta, \ N_2(t) \geqslant u(n)\right\}}_{=:Z(n)},$$

$$\text{(30)}$$

where $\bar{Y}_i(t) := \frac{\sum_{j=1}^{N_i(t)} Y_{i,j}}{N_i(t)}$ with $Y_{i,j} := X_{i,j} - \mu_i, \ i \in \{1, 2\}, j \in \mathbb{N}$ (Notice that these definitions of $\bar{Y}_i(t)$ and $Y_{i,j}$ are identical to their counterparts from the proof for arm 1.). From (30), it follows that

$$\mathbb{E}Z(n)$$

$$= \sum_{t=u(n)}^{n-1} \mathbb{P}\left(\bar{Y}_2(t) - \bar{Y}_1(t) \geqslant \sqrt{\rho \log t}\left(\frac{1}{\sqrt{N_1(t)}} - \frac{1}{\sqrt{N_2(t)}}\right) + \Delta, \ N_2(t) \geqslant u(n)\right)$$

$$= \sum_{t=u(n)}^{n-1} \mathbb{P}\left(\bar{Y}_2(t) - \bar{Y}_1(t) \geqslant \sqrt{\rho \log t}\left(\frac{1}{\sqrt{N_1(t)}} - \frac{1}{\sqrt{N_2(t)}}\right) + \sqrt{\frac{\theta \log n}{n}}, \ N_2(t) \geqslant u(n)\right)$$

$$\leqslant \sum_{t=u(n)}^{n-1} \mathbb{P}\left(\bar{Y}_2(t) - \bar{Y}_1(t) \geqslant \sqrt{\rho \log t}\left(\frac{1}{\sqrt{t - u(n)}} - \frac{1}{\sqrt{u(n)}}\right) + \sqrt{\frac{\theta \log n}{n}}\right)$$

$$\leqslant \sum_{t=u(n)}^{n-1} \mathbb{P}\left(\bar{Y}_2(t) - \bar{Y}_1(t) \geqslant \sqrt{\rho \log t}\left(\frac{1}{\sqrt{n - u(n)}} - \frac{1}{\sqrt{u(n)}}\right) + \sqrt{\frac{\theta \log t}{n}}\right)$$

$$\leqslant \sum_{t=u(n)}^{n-1} \mathbb{P}\left(\bar{Y}_2(t) - \bar{Y}_1(t) \geqslant \sqrt{\frac{\rho \log t}{n}}\left(\frac{1}{\sqrt{\lambda_\rho^*(\theta) - \epsilon}} - \frac{1}{\sqrt{1 - \lambda_\rho^*(\theta) + \epsilon}} + \sqrt{\frac{\theta}{\rho}}\right)\right),$$

where $\frac{1}{\sqrt{\lambda_\rho^*(\theta) - \epsilon}} - \frac{1}{\sqrt{1 - \lambda_\rho^*(\theta) + \epsilon}} + \sqrt{\frac{\theta}{\rho}} > 0$ is guaranteed since $\lambda_\rho^*(\theta)$ is the solution to (2). Also, $t \geqslant u(n) = \left\lceil \left(1 - \lambda_\rho^*(\theta) + \epsilon\right) n\right\rceil \implies n \leqslant \frac{t}{1 - \lambda_\rho^*(\theta) + \epsilon}$. Therefore,

$$\mathbb{E}Z(n)$$

$$\leqslant \sum_{t=u(n)}^{n-1} \mathbb{P}\left(\bar{Y}_2(t) - \bar{Y}_1(t) \geqslant \sqrt{1 - \lambda_\rho^*(\theta) + \epsilon}\sqrt{\frac{\rho \log t}{t}}\left(\frac{1}{\sqrt{\lambda_\rho^*(\theta) - \epsilon}} - \frac{1}{\sqrt{1 - \lambda_\rho^*(\theta) + \epsilon}} + \sqrt{\frac{\theta}{\rho}}\right)\right)$$

$$= \sum_{t=u(n)}^{n-1} \mathbb{P}\left(\underbrace{\sqrt{\frac{t}{\rho \log t}}\left(\bar{Y}_2(t) - \bar{Y}_1(t)\right)}_{=:W_t} \geqslant \underbrace{\sqrt{1 - \lambda_\rho^*(\theta) + \epsilon}\left(\frac{1}{\sqrt{\lambda_\rho^*(\theta) - \epsilon}} - \frac{1}{\sqrt{1 - \lambda_\rho^*(\theta) + \epsilon}} + \sqrt{\frac{\theta}{\rho}}\right)}_{=:\varepsilon(\theta, \rho, \epsilon) \ \left(\text{Note that } \varepsilon(\theta, \rho, \epsilon) > 0 \text{ for any } \epsilon \in \left(0, \lambda_\rho^*(\theta)\right)\right)}\right).$$

$$\text{(31)}$$

Recall that we have already handled $W_t$ (albeit a negated version thereof) in the proof for arm 1 in (25) and shown that $W_t \to 0$ almost surely in (28). Now consider an arbitrary $\delta > 0$. We then have

$$\mathbb{P}\left(N_2(n) - u(n) \geqslant \delta n\right) \leqslant \mathbb{P}\left(Z(n) \geqslant \delta n\right) \qquad \text{(using (30))}$$

$$\leqslant \frac{\mathbb{E}Z(n)}{\delta n} \qquad \text{(Markov's inequality)}$$

$$\leqslant \frac{1}{\delta n} \sum_{t=u(n)}^{n-1} \mathbb{P}\left(W_t \geqslant \varepsilon(\theta, \rho, \epsilon)\right) \qquad \text{(using (31))}$$

$$\leqslant \frac{1}{\delta} \sup_{t > \left(1 - \lambda_\rho^*(\theta)\right)n} \mathbb{P}\left(W_t \geqslant \varepsilon(\theta, \rho, \epsilon)\right). \qquad (32)$$

Taking limits on both sides of (32), we obtain

$$\limsup_{n \to \infty} \mathbb{P}\left(N_2(n) - u(n) \geqslant \delta n\right) \leqslant \frac{1}{\delta} \limsup_{n \to \infty} \mathbb{P}\left(W_n \geqslant \varepsilon(\theta, \rho, \epsilon)\right) = 0,$$

where the final conclusion follows since $W_n \to 0$ almost surely, and hence also in probability. Now since $u(n) = \left\lceil \left(1 - \lambda_\rho^*(\theta) + \epsilon\right) n \right\rceil$ and $\epsilon, \delta > 0$ are arbitrary, it follows that for any $\epsilon > 0$, we have $\lim_{n \to \infty} \mathbb{P}\left(\frac{N_2(n)}{n} \geqslant 1 - \lambda_\rho^*(\theta) + \epsilon\right) = 0$. From the proof for arm 1, we already know that $\lim_{n \to \infty} \mathbb{P}\left(\frac{N_2(n)}{n} \leqslant 1 - \lambda_\rho^*(\theta) - \epsilon\right) = 0$ holds for any $\epsilon > 0$. Therefore, it must be the case that $\frac{N_2(n)}{n} \xrightarrow[n \to \infty]{p} 1 - \lambda_\rho^*(\theta)$ and $\frac{N_1(n)}{n} \xrightarrow[n \to \infty]{p} \lambda_\rho^*(\theta)$, as desired. $\qquad \square$

## G  Proof of Theorem 2

### G.1  Proof of part (I)

Let $\theta_k, \tilde{\theta}_k$ be Beta$(1, k + 1)$-distributed, with $\theta_k, \tilde{\theta}_l$ independent $\forall\, k, l \in \mathbb{N} \cup \{0\}$. In the two-armed bandit with deterministic 0 rewards, at any time $n + 1$, the probability of playing arm 1 conditioned on the entire history up to that point, is given by $\mathbb{P}\left(\pi_{n+1} = 1 \mid \mathcal{F}_n\right) = \mathbb{P}\left(\theta_{N_1(n)} > \tilde{\theta}_{N_2(n)} \mid \mathcal{F}_n\right) = \frac{n - N_1(n) + 1}{n + 2}$ (using Fact (2)). Since the arms are identical, and $N_1(n) + N_2(n) = n$, we must have $\mathbb{E}\left(N_1(n)/n\right) = 1/2 \,\forall\, n \in \mathbb{N}$ by symmetry. Define $Z_n := N_1(n)/n$. Then, $Z_n$ evolves according to the following Markovian rule:

$$Z_{n+1} = \left(\frac{n}{n+1}\right) Z_n + \frac{Y\left(Z_n, n, \xi_n\right)}{n+1},$$

where $\{\xi_n\}$ is an independent noise process that is such that $Y\left(Z_n, n, \xi_n\right) | Z_n$ is distributed as Bernoulli$\left(\frac{n(1 - Z_n) + 1}{n+2}\right)$. Note that $Y(\cdot, \cdot, \cdot) \in \{0, 1\}$. Then,

$$Z_{n+1}^2 = \left(\frac{n}{n+1}\right)^2 Z_n^2 + \left(\frac{Y\left(Z_n, n, \xi_n\right)}{n+1}\right)^2 + \frac{2n Z_n Y\left(Z_n, n, \xi_n\right)}{(n+1)^2}$$

$$= \left(\frac{n}{n+1}\right)^2 Z_n^2 + \frac{Y\left(Z_n, n, \xi_n\right)}{(n+1)^2} + \frac{2n Z_n Y\left(Z_n, n, \xi_n\right)}{(n+1)^2}.$$

Solving the recursion for $Z_{n+1}^2$, we obtain

$$Z_{n+1}^2 = \frac{Z_1^2 + \sum_{t=1}^n \left[Y\left(Z_t, t, \xi_t\right) + 2t Z_t Y\left(Z_t, t, \xi_t\right)\right]}{(n+1)^2} = \frac{Z_1 + \sum_{t=1}^n \left[Y\left(Z_t, t, \xi_t\right) + 2t Z_t Y\left(Z_t, t, \xi_t\right)\right]}{(n+1)^2},$$

where the last equality follows since $Z_1 \in \{0, 1\}$. Taking expectations and using the fact that $\mathbb{E}Z_t = 1/2 \,\forall\, t \in \mathbb{N}$, yields

$$\mathbb{E}Z_{n+1}^2 = \frac{\frac{1}{2} + \sum_{t=1}^n \left(\frac{1}{2} + 2t\mathbb{E}\left[\frac{t Z_t(1 - Z_t) + Z_t}{t+2}\right]\right)}{(n+1)^2}.$$

Using $Z_t(1 - Z_t) \leqslant 1/4$, we get the relation

$$\mathbb{E}Z_{n+1}^2 \leqslant \frac{1 + \sum_{t=1}^{n}\left(1 + t\mathbb{E}\left[\frac{t+4Z_t}{t+2}\right]\right)}{2(n+1)^2} = \frac{n+1+\sum_{t=1}^{n}t}{2(n+1)^2} = \frac{n+2}{4(n+1)}.$$

Thus, $\text{Var}\left(\frac{N_1(n)}{n}\right) = \text{Var}(Z_n) \leqslant \frac{n+1}{4n} - \frac{1}{4} = \frac{1}{4n}$. Since $\mathbb{E}\left(\frac{N_1(n)}{n}\right) = \frac{1}{2}$, we conclude using Chebyshev's inequality that $\frac{N_1(n)}{n} \to \frac{1}{2}$ in probability as $n \to \infty$. $\qquad\square$

## G.2  Proof of part (II)

Our proof of this part is essentially pivoted on showing the stronger result that $\mathbb{P}(N_1(n) = m) = \frac{1}{n+1}$ for any $m \in \{0, ..., n\}$ and $n \in \mathbb{N}$. To this end, for an arbitrary $m$ in said interval, let $\mathrm{S}_m$ be the set of sample-paths of length $n$ such that $N_1(n) = m$ on each sample-path $\mathrm{s}_m \in \mathrm{S}_m$. Clearly, $|\mathrm{S}_m| = \binom{n}{m}$. Let $i(\mathrm{s}_m, t) \in \{1, 2\}$ denote the index of the arm pulled at time $t \in \{1, ..., n\}$ on $\mathrm{s}_m$, and let $\tilde{N}_j(t)$ denote the number of pulls of arm $j \in \{1, 2\}$ up to (and including) time $t$ on $\mathrm{s}_m$ (with $\tilde{N}_1(0) = \tilde{N}_2(0) := 0$). Note that $i(\mathrm{s}_m, t)$, $\tilde{N}_1(t)$ and $\tilde{N}_2(t)$ are deterministic for all $t \in \{1, ..., n\}$, once $\mathrm{s}_m$ is fixed. Let $\theta_k, \tilde{\theta}_k$ be Beta$(k+1, 1)$-distributed, with $\theta_k, \tilde{\theta}_l$ independent $\forall\, k, l \in \mathbb{N} \cup \{0\}$. It then follows that

$$\mathbb{P}(N_1(n) = m) = \sum_{\mathrm{s}_m \in \mathrm{S}_m} \prod_{t=1}^{n} \mathbb{P}\left(\theta_{\tilde{N}_{i(\mathrm{s}_m,t)}(t-1)} > \tilde{\theta}_{\tilde{N}_{\{1,2\}\setminus i(\mathrm{s}_m,t)}(t-1)}\right)$$

$$= \sum_{\mathrm{s}_m \in \mathrm{S}_m} \prod_{t=1}^{n} \left(\frac{\tilde{N}_{i(\mathrm{s}_m,t)}(t-1) + 1}{t+1}\right) \qquad \text{(using Fact (3))}$$

$$= \frac{1}{(n+1)!} \sum_{\mathrm{s}_m \in \mathrm{S}_m} \prod_{t=1}^{n} \left(\tilde{N}_{i(\mathrm{s}_m,t)}(t-1) + 1\right)$$

$$= \frac{1}{(n+1)!} \sum_{\mathrm{s}_m \in \mathrm{S}_m} m!(n-m)!,$$

where the last equality follows since $\tilde{N}_1(n) = m$, $\tilde{N}_2(n) = n - m$, $\tilde{N}_1(0) = \tilde{N}_2(0) = 0$ on $\mathrm{s}_m$. Therefore, we have for all $m \in \{0, ..., n\}$ that

$$\mathbb{P}(N_1(n) = m) = \frac{\binom{n}{m}m!(n-m)!}{(n+1)!} = \frac{n!}{(n+1)!} = \frac{1}{n+1}. \tag{33}$$

This, in fact, proves a stronger result that $N_1(n)/n$ is uniformly distributed on $\left\{0, \frac{1}{n}, \frac{2}{n}, ..., \frac{n-1}{n}, 1\right\}$ for any $n \in \mathbb{N}$. The desired result now follows as a corollary in the limit $n \to \infty$; for an arbitrary $x \in [0, 1]$, consider

$$\mathbb{P}\left(\frac{N_1(n)}{n} \leqslant x\right) = \sum_{m=0}^{\lfloor xn \rfloor} \mathbb{P}(N_1(n) = m) = \frac{\lfloor xn \rfloor + 1}{n+1},$$

where the last equality follows using (33). Thus, we have $\lim_{n\to\infty} \mathbb{P}\left(\frac{N_1(n)}{n} \leqslant x\right) = x$ for any $x \in [0, 1]$, i.e., $\frac{N_1(n)}{n}$ converges in law to the Uniform distribution on $[0, 1]$. $\qquad\square$

## H  Proof of Theorem 3

We essentially need to bound the growth rate of $R_n^\pi$ under the policy $\pi$ given by Algorithm 1 with $\rho > 1$, in three (exhaustive) regimes, viz., (i) $\Delta = o\left(\sqrt{\frac{\log n}{n}}\right)$ ("small gap"), (ii) $\Delta = \omega\left(\sqrt{\frac{\log n}{n}}\right)$ ("large gap"), and (iii) $\Delta = \Theta\left(\sqrt{\frac{\log n}{n}}\right)$ ("moderate gap"). We handle the three cases below separately.

### H.1 The "small gap" regime

Here, we have

$$\frac{\mathbb{E}R_n^\pi}{\sqrt{n\log n}} \leqslant \frac{\Delta n}{\sqrt{n\log n}} = \sqrt{\frac{\Delta^2 n}{\log n}}.$$

Since $\Delta = o\left(\sqrt{\frac{\log n}{n}}\right)$, it follows that $\mathbb{E}R_n^\pi = o\left(\sqrt{n\log n}\right)$. Therefore, we conclude using Markov's inequality that $R_n^\pi = o_p\left(\sqrt{n\log n}\right)$ whenever $\Delta = o\left(\sqrt{\frac{\log n}{n}}\right)$.

### H.2 The "large gap" regime

In this regime, we have

$$\frac{\mathbb{E}R_n^\pi}{\sqrt{n\log n}} \leqslant \frac{C\rho\left(\frac{\log n}{\Delta} + \frac{\Delta}{\rho-1}\right)}{\sqrt{n\log n}},$$

where $C$ is some absolute constant (follows from [6], Theorem 7). Since $\Delta = \omega\left(\sqrt{\frac{\log n}{n}}\right)$ and $\Delta \leqslant 1$ (rewards bounded in $[0,1]$), it follows that $\mathbb{E}R_n^\pi = o\left(\sqrt{n\log n}\right)$. Thus, we again conclude using Markov's inequality that $R_n^\pi = o_p\left(\sqrt{n\log n}\right)$ whenever $\Delta = \omega\left(\sqrt{\frac{\log n}{n}}\right)$.

### H.3 The "moderate gap" regime

Since $\Delta = \Theta\left(\sqrt{\frac{\log n}{n}}\right)$, there exists some $\theta \in \mathbb{R}_+$ and a diverging sequence of natural numbers $\{n_k\}_{k\in\mathbb{N}}$ such that $\Delta$ scales with the horizon of play $n_k$ along this sequence as $\Delta = \sqrt{\frac{\theta \log n_k}{n_k}}$. Without loss of generality, suppose that arm 1 is optimal, i.e., $\mu_1 \geqslant \mu_2$. We then have

$$\frac{R_{n_k}^\pi}{\sqrt{n_k\log n_k}} = \frac{\mu_1 n_k - \sum_{j=1}^{N_1(n_k)} X_{1,j} - \sum_{j=1}^{N_2(n_k)} X_{2,j}}{\sqrt{n_k\log n_k}} \qquad \text{(using (1))}$$

$$= \frac{\mu_1 n_k - \mu_1 N_1(n_k) - \mu_2 N_2(n_k) - \sum_{j=1}^{N_1(n_k)} Y_{1,j} - \sum_{j=1}^{N_2(n_k)} Y_{2,j}}{\sqrt{n_k\log n_k}},$$

where $Y_{i,j} := X_{i,j} - \mu_i$, $i \in \{1,2\}$, $j \in \mathbb{N}$. Therefore,

$$\frac{R_{n_k}^\pi}{\sqrt{n_k\log n_k}} = \frac{\Delta N_2(n_k) - \sum_{j=1}^{N_1(n_k)} Y_{1,j} - \sum_{j=1}^{N_2(n_k)} Y_{2,j}}{\sqrt{n_k\log n_k}}$$

$$= \Delta\sqrt{\frac{n_k}{\log n_k}}\left(\frac{N_2(n_k)}{n_k}\right) - \left(\frac{\sum_{j=1}^{N_1(n_k)} Y_{1,j} + \sum_{j=1}^{N_2(n_k)} Y_{2,j}}{\sqrt{n_k\log n_k}}\right)$$

$$= \sqrt{\theta}\left(\frac{N_2(n_k)}{n_k}\right) - \left(\frac{\sum_{j=1}^{N_1(n_k)} Y_{1,j} + \sum_{j=1}^{N_2(n_k)} Y_{2,j}}{\sqrt{n_k\log n_k}}\right). \qquad (34)$$

Consider the summation terms above. We have

$$\left|\frac{\sum_{j=1}^{N_1(n_k)} Y_{1,j} + \sum_{j=1}^{N_2(n_k)} Y_{2,j}}{\sqrt{n_k\log n_k}}\right| \leqslant \left|\frac{\sum_{j=1}^{N_1(n_k)} Y_{1,j}}{\sqrt{n_k\log n_k}}\right| + \left|\frac{\sum_{j=1}^{N_2(n_k)} Y_{2,j}}{\sqrt{n_k\log n_k}}\right|$$

$$\leqslant \left|\frac{\sum_{j=1}^{N_1(n_k)} Y_{1,j}}{\sqrt{N_1(n_k)\log N_1(n_k)}}\right| + \left|\frac{\sum_{j=1}^{N_2(n_k)} Y_{2,j}}{\sqrt{N_2(n_k)\log N_2(n_k)}}\right|. \qquad (35)$$

Since $N_i(n_k)$, for each $i \in \{1, 2\}$, can be lower-bounded *path-wise* by a deterministic monotone increasing coercive function of $k$, say $f(k)$, we have

$$\left| \frac{\sum_{j=1}^{N_i(n_k)} Y_{i,j}}{\sqrt{N_i(n_k) \log N_i(n_k)}} \right| \leqslant \sup_{m \geqslant f(k)} \left| \frac{\sum_{j=1}^{m} Y_{i,j}}{\sqrt{m \log m}} \right| \quad \forall\, i \in \{1, 2\}. \tag{36}$$

Since $Y_{i,j}$'s are independent, zero-mean, bounded random variables, it follows from the Law of the Iterated Logarithm (see [11], Theorem 8.5.2) that

$$\sup_{m \geqslant f(k)} \left| \frac{\sum_{j=1}^{m} Y_{i,j}}{\sqrt{m \log m}} \right| \xrightarrow[k \to \infty]{\text{w.p. } 1} 0 \quad \forall\, i \in \{1, 2\}. \tag{37}$$

Combining (34), (35), (36) and (37), we conclude

$$\frac{R_{n_k}^{\pi}}{\sqrt{n_k \log n_k}} = \sqrt{\theta}\left( \frac{N_2(n_k)}{n_k} \right) + o_p(1).$$

From Theorem 1, we know that when $\Delta \sim \sqrt{\frac{\theta \log n_k}{n_k}}$, $\frac{N_2(n_k)}{n_k} \xrightarrow[k \to \infty]{p} 1 - \lambda_\rho^*(\theta)$. Thus, it follows that

$$\frac{R_{n_k}^{\pi}}{\sqrt{n_k \log n_k}} \xrightarrow[k \to \infty]{p} \sqrt{\theta}\left(1 - \lambda_\rho^*(\theta)\right) = h_\rho(\theta).$$

Since $\theta \in \mathbb{R}_+$ is arbitrary, the worst-case regret in the $\Delta = \Theta\left(\sqrt{\frac{\log n}{n}}\right)$ regime corresponds to the choice of $\theta$ given by $\theta_\rho^* = \arg\max_{\theta \geqslant 0} h_\rho(\theta)$. Since we already know that $R_n^{\pi} = o_p\left(\sqrt{n \log n}\right)$ in the other two regimes ("small" and "large" gaps), it must be that the $\theta_\rho^*$ so obtained indeed corresponds to the global (in $\Delta$) worst-case regret of Algorithm 1. $\qquad \square$

# I   Proof of Theorem 4

**Notation.** Let $\mathcal{C}$ be the space of continuous functions $[0, 1] \mapsto \mathbb{R}^2$, endowed with the uniform metric. Let $\mathcal{D}$ be the space of right-continuous functions with left limits, mapping $[0, 1] \mapsto \mathbb{R}^2$, and endowed with the Skorohod metric (see [8], Chapters 2 and 3, for an overview). Let $\mathcal{D}_0$ be the set of elements of $\mathcal{D}$ of the form $(\phi_1, \phi_2)$, where $\phi_i$ is a non-decreasing real-valued function satisfying $0 \leqslant \phi_i(t) \leqslant 1$ for $i \in \{1, 2\}$ and $t \in [0, 1]$. For $t \in [0, 1]$, denote the identity map by $\mathfrak{e}(t) := t$.

For $i \in \{1, 2\}$ and $t \in [0, 1]$, define $\Psi_{i,n}(t) := \frac{\sum_{j=1}^{\lfloor nt \rfloor} X_{i,j} - \mu n t}{\sqrt{n}}$. Then, $(\Psi_{1,n}, \Psi_{2,n}) \in \mathcal{D}$. Also for $i \in \{1, 2\}$ and $t \in [0, 1]$, define $W_i(t) := \theta_i t + \sigma_i B_i'(t)$, where $B_1'$ and $B_2'$ are independent standard Brownian motions in $\mathbb{R}$. Note that $\mathbb{P}(W_i \in \mathcal{C}) = 1$ for $i \in \{1, 2\}$. Since $(X_{i,j})_{i \in \{1,2\}, j \in \mathbb{N}}$'s are independent random variables (i.i.d. within and independent across sequences), we know from Donsker's Theorem (see [8], Section 14, for details) that as $n \to \infty$,

$$(\Psi_{1,n}, \Psi_{2,n}) \Rightarrow (W_1, W_2) \text{ in } \mathcal{D}.$$

For $i \in \{1, 2\}$ and $t \in [0, 1]$, define $\phi_{i,n}(t) := \frac{N_i(\lfloor nt \rfloor)}{n}$. Thus, $(\phi_{1,n}, \phi_{2,n}) \in \mathcal{D}_0$, and it follows from the result for the "small gap" regime in Theorem 1 that as $n \to \infty$,

$$(\phi_{1,n}, \phi_{2,n}) \xrightarrow{p} \left( \frac{\mathfrak{e}}{2}, \frac{\mathfrak{e}}{2} \right) \text{ in } \mathcal{D}_0.$$

Thus, we have convergence in the product space (see [8], Theorem 3.9), i.e., as $n \to \infty$,

$$(\Psi_{1,n}, \Psi_{2,n}, \phi_{1,n}, \phi_{2,n}) \Rightarrow \left( W_1, W_2, \frac{\mathfrak{e}}{2}, \frac{\mathfrak{e}}{2} \right) \text{ in } \mathcal{D} \times \mathcal{D}_0.$$

For $i \in \{1, 2\}$ and $t \in [0, 1]$, define the composition $(\Psi_{i,n} \circ \phi_{i,n})(t) := \Psi_{i,n}(\phi_{i,n}(t))$, and $\left(W_i \circ \frac{\mathfrak{e}}{2}\right)(t) := W_i\left(\frac{\mathfrak{e}(t)}{2}\right) = W_i\left(\frac{t}{2}\right)$. Since $W_1, W_2, \mathfrak{e} \in \mathcal{C}$ w.p. 1, it follows from the *random time-change lemma* (see [8], Section 14, for details) that as $n \to \infty$

$$(\Psi_{1,n} \circ \phi_{1,n}, \Psi_{2,n} \circ \phi_{2,n}) \Rightarrow \left( W_1 \circ \frac{\mathfrak{e}}{2}, W_2 \circ \frac{\mathfrak{e}}{2} \right) \text{ in } \mathcal{D}.$$

The stated assertion on cumulative rewards now follows by recognizing for $i \in \{1, 2\}$ and $t \in [0, 1]$ that $(\Psi_{i,n} \circ \phi_{i,n})(t) = \frac{\tilde{S}_{i,\lfloor nt \rfloor}}{\sqrt{n}}$, and defining $B_i(t) := \sqrt{2} B_i'\left(\frac{t}{2}\right)$. To prove the assertion on regret, assume without loss of generality that arm 1 is optimal, i.e., $\theta_1 \geqslant \theta_2$. Then, the result follows after a direct application of the Continuous Mapping Theorem (see [8], Theorem 2.7), to wit,

$$R_{\lfloor nt \rfloor}^{\pi} = \left(\mu + \frac{\theta_1}{\sqrt{n}}\right) \lfloor nt \rfloor - S_{1,\lfloor nt \rfloor} - S_{2,\lfloor nt \rfloor} = \frac{\theta_1 \lfloor nt \rfloor}{\sqrt{n}} - \left(\tilde{S}_{1,\lfloor nt \rfloor} + \tilde{S}_{2,\lfloor nt \rfloor}\right),$$

and therefore as $n \to \infty$,

$$\left(\frac{R_{\lfloor nt \rfloor}^{\pi}}{\sqrt{n}}\right)_{t \in [0,1]} \Rightarrow \left(\theta_1 t - \left(\left(\frac{\theta_1 + \theta_2}{2}\right) t + \frac{\sigma_1}{\sqrt{2}} B_1(t) + \frac{\sigma_2}{\sqrt{2}} B_2(t)\right)\right)_{t \in [0,1]} = \left(\frac{\Delta_0 t}{2} + \sqrt{\frac{\sigma_1^2 + \sigma_2^2}{2}} \tilde{B}(t)\right)_{t \in [0,1]},$$

where $\tilde{B}(t) := -\sqrt{\frac{\sigma_1^2}{\sigma_1^2 + \sigma_2^2}} B_1(t) - \sqrt{\frac{\sigma_2^2}{\sigma_1^2 + \sigma_2^2}} B_2(t)$. $\qquad\square$

# J Proof of Theorem 5

We will prove this result in two parts; the preamble in J.1 below will prove a meta-result stating that $N_i(n)/n > 1/(2|\mathcal{I}|)$ with high probability (approaching 1 as $n \to \infty$) for any arm $i \in \mathcal{I}$. We will then leverage this meta-result to prove the assertions of the theorem in J.2.

## J.1 Preamble

Let $L := |\mathcal{I}|$. If $L = 1$, the result follows trivially from the standard logarithmic bound for the expected regret (Theorem 7 in [6]), followed by Markov's inequality. Therefore, without loss of generality, suppose that $|\mathcal{I}| \geqslant 2$, and fix an arbitrary arm $i \in \mathcal{I}$. Then, we know that the following is true for any integer $u > 1$:

$$N_i(n) \leqslant u + \sum_{t=u}^{n-1} \mathbb{1}\{\pi_{t+1} = i, \ N_i(t) \geqslant u\}$$

$$\leqslant u + \sum_{t=u}^{n-1} \mathbb{1}\left\{\pi_{t+1} = i, \ N_i(t) \geqslant u, \ \sum_{j \in \mathcal{I}\setminus\{i\}} N_j(t) \leqslant t - u\right\},$$

where $\pi_{t+1} \in [K]$ indicates the arm played at time $t + 1$. In particular, the above holds also for $u = \left\lceil \left(\frac{1}{L} + \epsilon\right) n \right\rceil$, where $\epsilon \in \left(0, \frac{L-1}{L}\right)$ is arbitrarily chosen. We will fix this $u$ going forward, even though we may not always express its value explicitly for readability of the analysis that follows. We thus have

$$N_i(n) \leqslant u + \sum_{t=u}^{n-1} \mathbb{1}\left\{B_{i,N_i(t),t} \geqslant \max_{\hat{j} \in [K]\setminus\{i\}} B_{\hat{j},N_{\hat{j}}(t),t}, \ N_i(t) \geqslant u, \ \sum_{j \in \mathcal{I}\setminus\{i\}} N_j(t) \leqslant t - u\right\}$$

$$\leqslant u + \sum_{t=u}^{n-1} \mathbb{1}\left\{B_{i,N_i(t),t} \geqslant \max_{\hat{j} \in \mathcal{I}\setminus\{i\}} B_{\hat{j},N_{\hat{j}}(t),t}, \ N_i(t) \geqslant u, \ \sum_{j \in \mathcal{I}\setminus\{i\}} N_j(t) \leqslant t - u\right\}, \quad (38)$$

where $B_{k,s,t} := \hat{X}_k(s) + \sqrt{(\rho \log t)/s}$ for $k \in [K]$, and $\hat{X}_k(s) := \sum_{l=1}^{s} X_{k,l}/s$ denotes the empirical mean reward from the "first $s$ plays" of arm $k$ (Note the distinction from $\bar{X}_k(s)$, which has been defined before as the empirical mean reward of arm $k$ "at time $s$," i.e., mean over its "first $N_k(s)$ plays"). Now observe that

$$\left\{\sum_{j \in \mathcal{I}\setminus\{i\}} N_j(t) \leqslant t - u\right\} \subseteq \left\{\exists j \in \mathcal{I}\setminus\{i\} : N_j(t) \leqslant \frac{t-u}{L-1}\right\} \subseteq \left\{\exists j \in \mathcal{I}\setminus\{i\} : N_j(t) \leqslant \left(\frac{1}{L} - \frac{\epsilon}{L-1}\right) t\right\},$$

$$(39)$$

where the last inclusion follows using $u = \lceil \left( \frac{1}{L} + \epsilon \right) n \rceil$ and $n \geq t$. Combining (38) and (39) using the Union bound, we obtain

$$N_i(n) \leq u + \sum_{t=u}^{n-1} \sum_{j \in \mathcal{I} \setminus \{i\}} \mathbb{1} \left\{ B_{i,N_i(t),t} \geq \max_{\hat{j} \in \mathcal{I} \setminus \{i\}} B_{\hat{j},N_{\hat{j}}(t),t}, \; N_i(t) \geq u, \; N_j(t) \leq \left( \frac{1}{L} - \frac{\epsilon}{L-1} \right) t \right\}$$

$$\leq u + \sum_{t=u}^{n-1} \sum_{j \in \mathcal{I} \setminus \{i\}} \mathbb{1} \left\{ B_{i,N_i(t),t} \geq B_{j,N_j(t),t}, \; N_i(t) \geq u, \; N_j(t) \leq \left( \frac{1}{L} - \frac{\epsilon}{L-1} \right) t \right\}$$

$$\leq u + \underbrace{\sum_{t=u}^{n-1} \sum_{j \in \mathcal{I} \setminus \{i\}} \mathbb{1} \left\{ B_{i,N_i(t),t} \geq B_{j,N_j(t),t}, \; N_i(t) \geq \left( \frac{1}{L} + \epsilon \right) t, \; N_j(t) \leq \left( \frac{1}{L} - \frac{\epsilon}{L-1} \right) t \right\}}_{=:Z_n},$$

$$\tag{40}$$

where the last inequality again uses $u = \lceil \left( \frac{1}{L} + \epsilon \right) n \rceil$ and $n \geq t$. Define the events: $E_i := \left\{ N_i(t) \geq \left( \frac{1}{L} + \epsilon \right) t \right\}$, and $E_j := \left\{ N_j(t) \leq \left( \frac{1}{L} - \frac{\epsilon}{L-1} \right) t \right\}$. Now,

$$\mathbb{E} Z_n$$

$$= \sum_{t=u}^{n-1} \sum_{j \in \mathcal{I} \setminus \{i\}} \mathbb{P} \left( B_{i,N_i(t),t} \geq B_{j,N_j(t),t}, \; E_i, \; E_j \right)$$

$$= \sum_{t=u}^{n-1} \sum_{j \in \mathcal{I} \setminus \{i\}} \mathbb{P} \left( \hat{Y}_i \left( N_i(t) \right) - \hat{Y}_j \left( N_j(t) \right) \geq \sqrt{\rho \log t} \left( \frac{1}{\sqrt{N_j(t)}} - \frac{1}{\sqrt{N_i(t)}} \right), \; E_i, \; E_j \right), \quad \text{(41)}$$

where $\hat{Y}_k(s) := \sum_{l=1}^{s} Y_{k,l}/s$ and $Y_{k,l} := X_{k,l} - \mathbb{E} X_{k,l}$ for $k \in [K]$, $s \in \mathbb{N}$, $l \in \mathbb{N}$. The last equality above follows since $i, j \in \mathcal{I}$ and the mean rewards of arms in $\mathcal{I}$ are equal. Thus,

$$\mathbb{E} Z_n \leq \sum_{t=u}^{n-1} \sum_{j \in \mathcal{I} \setminus \{i\}} \sum_{m_i = \lceil \left( \frac{1}{L} + \epsilon \right) t \rceil}^{t} \sum_{m_j = 1}^{\lfloor \left( \frac{1}{L} - \frac{\epsilon}{L-1} \right) t \rfloor} \mathbb{P} \left( \hat{Y}_i \left( m_i \right) - \hat{Y}_j \left( m_j \right) \geq \sqrt{\rho \log t} \left( \frac{1}{\sqrt{m_j}} - \frac{1}{\sqrt{m_i}} \right) \right).$$

Since $\mathbb{E} \left[ \hat{Y}_i \left( m_i \right) - \hat{Y}_j \left( m_j \right) \right] = 0$, and $m_j < m_i$ over the range of the summation above, we can use the Chernoff-Hoeffding bound (Fact 1) to obtain

$$\mathbb{E} Z_n \leq \sum_{t=u}^{n-1} \sum_{j \in \mathcal{I} \setminus \{i\}} \sum_{m_i = \lceil \left( \frac{1}{L} + \epsilon \right) t \rceil}^{t} \sum_{m_j = 1}^{\lfloor \left( \frac{1}{L} - \frac{\epsilon}{L-1} \right) t \rfloor} \exp \left[ -2\rho \left( 1 - 2 \sqrt{\frac{m_i m_j}{(m_i + m_j)^2}} \right) \log t \right]. \quad \text{(42)}$$

Let $\gamma := m_i/(m_i + m_j)$. Then, $\gamma \geq \frac{\frac{1}{L} + \epsilon}{\frac{2}{L} + \left( \frac{L-2}{L-1} \right) \epsilon} > 1/2$ over the range of the summation in (42). Consequently, $\gamma(1 - \gamma)$ is maximized at $\gamma = \frac{\frac{1}{L} + \epsilon}{\frac{2}{L} + \left( \frac{L-2}{L-1} \right) \epsilon}$, and therefore, $m_i m_j / \left( m_i + m_j \right)^2 = \gamma(1 - \gamma) \leq \left( f(\epsilon, L) \right)^2 < 1/4$ in (42), where $f(\epsilon, L)$ as defined as:

$$f(\epsilon, L) := \sqrt{\frac{(L-1)(1 + \epsilon L)\left( L - 1 - \epsilon L \right)}{\left( 2(L-1) + L(L-2)\epsilon \right)^2}}. \quad \text{(43)}$$

Combining (42) and (43), we obtain

$$\mathbb{E} Z_n \leq (L-1) \sum_{t=u}^{n-1} t^{-2(\rho - 1 - 2\rho f(\epsilon, L))}. \quad \text{(44)}$$

Now consider an arbitrary $\delta > 0$. From (40), we have

$$\mathbb{P} \left( N_i(n) \geq u + \delta n \right) \leq \mathbb{P} \left( Z_n \geq \delta n \right) \underset{(\star)}{\leq} \frac{\mathbb{E} Z_n}{\delta n} \underset{(\dagger)}{\leq} \left( \frac{L-1}{\delta n} \right) \sum_{t=u}^{n-1} t^{-2(\rho(1 - 2f(\epsilon, L)) - 1)}, \quad \text{(45)}$$

where $(\star)$ is due to Markov's inequality, and $(\dagger)$ follows using (44). Observe from (43) that $f(\epsilon, L)$ is monotone decreasing in $\epsilon$ over the interval $\epsilon \in \left[0, \frac{L-1}{L}\right]$, with $f(0, L) = 1/2$ and $f\left(\frac{L-1}{L}, L\right) = 0$. Therefore, $1 - 2f(\epsilon, L) > 0$ in the interval $\epsilon \in \left(0, \frac{L-1}{L}\right]$. Thus, for $\rho$ large enough, the exponent of $t$ in (45) can be made arbitrarily small. That is, $\exists \; \rho_0 \in \mathbb{R}_+$ s.t. for all $\rho \geqslant \rho_0$, we have $2\left(\rho\left(1 - 2f(\epsilon, L)\right) - 1\right) > 0 \; \forall \; \epsilon \in \left[\frac{1}{2L(L-1)}, \frac{L-1}{L}\right]$. Now supposing $\rho \geqslant \rho_0$, plug in $\epsilon = \frac{1}{2L(L-1)}$ in (45) (this includes substituting $u = \left\lceil \left(\frac{1}{L} + \frac{1}{2L(L-1)}\right) n \right\rceil$). Then since $\delta > 0$ and $i \in \mathcal{I}$ are arbitrary, it follows that for any $\delta > 0$ and $i \in \mathcal{I}$,

$$\lim_{n \to \infty} \mathbb{P}\left(N_i(n) \geqslant \left(\frac{1}{L} + \frac{1}{2L(L-1)} + \delta\right) n\right) = \frac{L^{2\rho}}{\delta} \lim_{n \to \infty} n^{-2\left(\rho\left(1 - 2f\left(\frac{1}{2L(L-1)}, L\right)\right) - 1\right)} = 0.$$

$$(46)$$

Notice that for any $\delta > 0$ and $i \in \mathcal{I}$,

$$\mathbb{P}\left(N_i(n) + \sum_{j \in [K] \setminus \mathcal{I}} N_j(n) \leqslant \left(\frac{1}{2L} - (L-1)\delta\right) n\right) = \mathbb{P}\left(\sum_{j \in \mathcal{I} \setminus \{i\}} N_j(n) \geqslant (L-1)\left(\frac{1}{L} + \frac{1}{2L(L-1)} + \delta\right) n\right)$$

$$\leqslant \sum_{j \in \mathcal{I} \setminus \{i\}} \mathbb{P}\left(N_j(n) \geqslant \left(\frac{1}{L} + \frac{1}{2L(L-1)} + \delta\right) n\right),$$

where the last inequality follows using the Union bound. Taking limits on both sides above, we conclude using (46) that for any $\delta > 0$ and $i \in \mathcal{I}$,

$$\lim_{n \to \infty} \mathbb{P}\left(N_i(n) + \sum_{j \in [K] \setminus \mathcal{I}} N_j(n) \leqslant \left(\frac{1}{2L} - (L-1)\delta\right) n\right) = 0. \qquad (47)$$

If $\mathcal{I} = [K]$, the conclusion that $N_i(n)/n > 1/(2L) = 1/(2K)$ with high probability (approaching 1 as $n \to \infty$) for all $i \in [K]$, is immediate from (47). If $\mathcal{I} \neq [K]$, then $\sum_{j \in [K] \setminus \mathcal{I}} \mathbb{E}\left(N_j(n)/n\right) \leqslant CK\rho\left[\left(\frac{1}{\Delta_{\min}^2}\right)\left(\frac{\log n}{n}\right) + \frac{1}{(\rho-1)n}\right]$ for some absolute constant $C > 0$ follows from [6], Theorem 7. Consequently if $\Delta_{\min} = \omega\left(\sqrt{\frac{\log n}{n}}\right)$, Markov's inequality implies that $\sum_{j \in [K] \setminus \mathcal{I}} N_j(n)/n = o_p(1)$. Thus, it again follows using (47) that $N_i(n)/n > 1/(2L)$ with high probability (approaching 1 as $n \to \infty$) for all $i \in \mathcal{I}$. $\qquad \square$

## J.2   Proof of part (I) and (II)

Note that the following holds for any integer $u > 1$ and any arm $i \in [K]$:

$$N_i(n) \leqslant u + \sum_{t=u+1}^{n} \mathbb{1}\left\{\pi_t = i, \; N_i(t-1) \geqslant u\right\},$$

where $\pi_t \in [K]$ indicates the arm played at time $t$. In particular, the above is true also for $u = N_j(n) + \lceil \epsilon n \rceil$, where $j \in [K] \setminus \{i\}$ and $\epsilon > 0$ are arbitrarily chosen. Without loss of generality, suppose that $|\mathcal{I}| \geqslant 2$ (the result is trivial for $|\mathcal{I}| = 1$), and fix two arbitrary arms $i, j \in \mathcal{I}$. Then,

$$N_i(n) \leqslant N_j(n) + \lceil \epsilon n \rceil + \sum_{t=N_j(n)+\lceil \epsilon n \rceil+1}^{n} \mathbb{1}\left\{\pi_t = i, \; N_i(t-1) \geqslant N_j(n) + \lceil \epsilon n \rceil\right\}$$

$$\leqslant N_j(n) + \lceil \epsilon n \rceil + \sum_{t=\lceil \epsilon n \rceil+1}^{n} \mathbb{1}\left\{\pi_t = i, \; N_i(t-1) \geqslant N_j(n) + \epsilon n\right\}$$

$$\leqslant N_j(n) + \lceil \epsilon n \rceil + \sum_{t=\lceil \epsilon n \rceil+1}^{n} \mathbb{1}\left\{B_{i,N_i(t-1),t-1} \geqslant B_{j,N_j(t-1),t-1}, \; N_i(t-1) \geqslant N_j(n) + \epsilon n\right\},$$

where $B_{k,s,t} := \hat{X}_k(s) + \sqrt{(\rho \log t)/s}$ for $k \in [K]$, with $\hat{X}_k(s)$ denoting the empirical mean reward from the first $s$ plays of arm $k$. Then,

$$N_i(n) \leqslant N_j(n) + \lceil \epsilon n \rceil + \sum_{t=\lceil \epsilon n \rceil}^{n-1} \mathbb{1}\left\{ B_{i,N_i(t),t} \geqslant B_{j,N_j(t),t}, \ N_i(t) \geqslant N_j(n) + \epsilon n \right\}$$

$$\leqslant N_j(n) + \lceil \epsilon n \rceil + \sum_{t=\lceil \epsilon n \rceil}^{n-1} \mathbb{1}\left\{ B_{i,N_i(t),t} \geqslant B_{j,N_j(t),t}, \ N_i(t) \geqslant N_j(t) + \epsilon t \right\}$$

$$\leqslant N_j(n) + \epsilon n + 1 + Z_n, \tag{48}$$

where $Z_n := \sum_{t=\lceil \epsilon n \rceil}^{n-1} \mathbb{1}\left\{ B_{i,N_i(t),t} \geqslant B_{j,N_j(t),t}, \ N_i(t) \geqslant N_j(t) + \epsilon t \right\}$. Now,

$$\mathbb{E}Z_n$$

$$= \sum_{t=\lceil \epsilon n \rceil}^{n-1} \mathbb{P}\left\{ B_{i,N_i(t),t} \geqslant B_{j,N_j(t),t}, \ N_i(t) \geqslant N_j(t) + \epsilon t \right\}$$

$$= \sum_{t=\lceil \epsilon n \rceil}^{n-1} \mathbb{P}\left\{ \hat{X}_i(N_i(t)) - \hat{X}_j(N_j(t)) \geqslant \sqrt{\rho \log t}\left( \frac{1}{\sqrt{N_j(t)}} - \frac{1}{\sqrt{N_i(t)}} \right), \ N_i(t) \geqslant N_j(t) + \epsilon t \right\}$$

$$= \sum_{t=\lceil \epsilon n \rceil}^{n-1} \mathbb{P}\left\{ \hat{Y}_i(N_i(t)) - \hat{Y}_j(N_j(t)) \geqslant \sqrt{\rho \log t}\left( \frac{1}{\sqrt{N_j(t)}} - \frac{1}{\sqrt{N_i(t)}} \right), \ N_i(t) \geqslant N_j(t) + \epsilon t \right\},$$

where $\hat{Y}_k(s) := \sum_{l=1}^{s} Y_{k,l}/s$ for $k \in [K]$, $s \in \mathbb{N}$, with $Y_{k,l} := X_{k,l} - \mathbb{E}X_{k,l}$ for $l \in \mathbb{N}$. The last equality above follows since $i, j \in \mathcal{I}$ and the mean rewards of arms in $\mathcal{I}$ are equal. Thus,

$$\mathbb{E}Z_n$$

$$= \sum_{t=\lceil \epsilon n \rceil}^{n-1} \mathbb{P}\left( \hat{Y}_i(N_i(t)) - \hat{Y}_j(N_j(t)) \geqslant \sqrt{\frac{\rho \log t}{N_i(t)}}\left( \sqrt{\frac{N_i(t)}{N_j(t)}} - 1 \right), \ N_i(t) \geqslant N_j(t) + \epsilon t \right)$$

$$\leqslant \sum_{t=\lceil \epsilon n \rceil}^{n-1} \mathbb{P}\left( \hat{Y}_i(N_i(t)) - \hat{Y}_j(N_j(t)) \geqslant \sqrt{\frac{\rho \log t}{t}}\left( \sqrt{1+\epsilon} - 1 \right), \ N_i(t) \geqslant N_j(t) + \epsilon t \right)$$

$$\leqslant \sum_{t=\lceil \epsilon n \rceil}^{n-1} \mathbb{P}\left( W_t \geqslant \sqrt{1+\epsilon} - 1 \right), \tag{49}$$

where $W_t := \sqrt{\frac{t}{\rho \log t}}\left( \frac{\sum_{l=1}^{N_i(t)} Y_{i,l}}{N_i(t)} - \frac{\sum_{l=1}^{N_j(t)} Y_{j,l}}{N_j(t)} \right)$. Now,

$$|W_t|$$

$$\leqslant \sqrt{\frac{t}{\rho \log t}}\left( \left| \frac{\sum_{l=1}^{N_i(t)} Y_{i,l}}{N_i(t)} \right| + \left| \frac{\sum_{l=1}^{N_j(t)} Y_{j,l}}{N_j(t)} \right| \right)$$

$$= \sqrt{\frac{2t}{\rho \log t}}\left( \sqrt{\frac{\log \log N_i(t)}{N_i(t)}} \left| \frac{\sum_{l=1}^{N_i(t)} Y_{i,l}}{\sqrt{2N_i(t) \log \log N_i(t)}} \right| + \sqrt{\frac{\log \log N_j(t)}{N_j(t)}} \left| \frac{\sum_{l=1}^{N_j(t)} Y_{j,j}}{\sqrt{2N_j(t) \log \log N_j(t)}} \right| \right)$$

$$\leqslant \sqrt{\frac{2t}{\rho \log t}}\left( \sqrt{\frac{\log \log t}{N_i(t)}} \left| \frac{\sum_{l=1}^{N_i(t)} Y_{i,l}}{\sqrt{2N_i(t) \log \log N_i(t)}} \right| + \sqrt{\frac{\log \log t}{N_j(t)}} \left| \frac{\sum_{l=1}^{N_j(t)} Y_{j,l}}{\sqrt{2N_j(t) \log \log N_j(t)}} \right| \right)$$

$$= \sqrt{\frac{2 \log \log t}{\rho \log t}}\left( \sqrt{\frac{t}{N_i(t)}} \left| \frac{\sum_{l=1}^{N_i(t)} Y_{i,l}}{\sqrt{2N_i(t) \log \log N_i(t)}} \right| + \sqrt{\frac{t}{N_j(t)}} \left| \frac{\sum_{l=1}^{N_j(t)} Y_{j,l}}{\sqrt{2N_j(t) \log \log N_j(t)}} \right| \right). \tag{50}$$

We know that $N_k(t)$, for any arm $k \in [K]$, can be lower-bounded *path-wise* by a deterministic monotone increasing *coercive* function of $t$, say $h(t)$. This follows as a trivial consequence of the

structure of the policy, and the fact that the rewards are uniformly bounded. Therefore, we have for any arm $k \in \mathcal{I}$ that

$$\left| \frac{\sum_{l=1}^{N_k(t)} Y_{k,l}}{\sqrt{2N_k(t) \log \log N_k(t)}} \right| \leqslant \sup_{m \geqslant h(t)} \left| \frac{\sum_{l=1}^{m} Y_{k,l}}{\sqrt{2m \log \log m}} \right|$$

$$\implies \limsup_{t \to \infty} \left| \frac{\sum_{l=1}^{N_k(t)} Y_{k,l}}{\sqrt{2N_k(t) \log \log N_k(t)}} \right| \leqslant \limsup_{t \to \infty} \left| \frac{\sum_{l=1}^{t} Y_{k,l}}{\sqrt{2t \log \log t}} \right|. \tag{51}$$

For any $k \in \mathcal{I}$, we know that $\{Y_{k,l} : l \in \mathbb{N}\}$ is a collection of i.i.d. random variables with $\mathbb{E}Y_{k,1} = 0$ and $\mathrm{Var}\,(Y_{k,1}) = \mathrm{Var}\,(X_{k,1}) \leqslant 1$. Therefore, we conclude using the Law of the Iterated Logarithm (see Theorem 8.5.2 in [11]) in (51) that

$$\limsup_{t \to \infty} \left| \frac{\sum_{l=1}^{N_k(t)} Y_{k,l}}{\sqrt{2N_k(t) \log \log N_k(t)}} \right| \leqslant 1 \quad \text{w.p. } 1 \ \forall\, k \in \mathcal{I}. \tag{52}$$

Using (50), (52), and the meta-result from the preamble in J.1 that $N_k(t)/t > 1/\left(2\,|\mathcal{I}|\right)$ with high probability (approaching 1 as $t \to \infty$) for any arm $k \in \mathcal{I}$, we conclude that

$$W_t \overset{p}{\to} 0 \quad \text{as } t \to \infty. \tag{53}$$

Now,

$$\mathbb{P}\left( \frac{N_i(n) - N_j(n)}{n} \geqslant 2\epsilon \right) \underset{(\dagger)}{\leqslant} \mathbb{P}(1 + Z_n \geqslant \epsilon n) \underset{(\ddagger)}{\leqslant} \frac{1 + \mathbb{E}Z_n}{\epsilon n} \underset{(\star)}{\leqslant} \frac{1}{\epsilon n} + \frac{1}{\epsilon n} \sum_{t=\lceil \epsilon n \rceil}^{n-1} \mathbb{P}\left( W_t > \sqrt{1+\epsilon} - 1 \right),$$

where ($\dagger$) follows using (48), ($\ddagger$) using Markov's inequality, and ($\star$) from (49). Therefore,

$$\mathbb{P}\left( \frac{N_i(n) - N_j(n)}{n} \geqslant 2\epsilon \right) \leqslant \frac{1}{\epsilon n} + \left( \frac{1-\epsilon}{\epsilon} \right) \sup_{t \geqslant \epsilon n} \mathbb{P}\left( W_t > \sqrt{1+\epsilon} - 1 \right). \tag{54}$$

Since $\epsilon > 0$ is arbitrary, we conclude using (53) and (54) that for any $\epsilon > 0$,

$$\lim_{n \to \infty} \mathbb{P}\left( \frac{N_i(n) - N_j(n)}{n} \geqslant 2\epsilon \right) = 0. \tag{55}$$

Our proof is symmetric w.r.t. the labels $i, j$, therefore, an identical result holds also with the labels interchanged in (55). Thus, we have $N_i(n)/n - N_j(n)/n \overset{p}{\to} 0$. Since $i, j$ are arbitrary in $\mathcal{I}$, the aforementioned convergence holds for any pair of arms in $\mathcal{I}$. Now if $\mathcal{I} = [K]$, we are done. If $\mathcal{I} \neq [K]$, then $\sum_{i \in [K] \setminus \mathcal{I}} \mathbb{E}\left( N_i(n)/n \right) \leqslant CK\rho \left[ \left( \frac{1}{\Delta_{\min}^2} \right) \left( \frac{\log n}{n} \right) + \frac{1}{(\rho-1)n} \right]$ for some absolute constant $C > 0$ follows from Theorem 7 in [6]. Consequently if $\Delta_{\min} = \omega\left( \sqrt{\frac{\log n}{n}} \right)$, it would follow from Markov's inequality that $\sum_{i \in [K] \setminus \mathcal{I}} N_i(n)/n = o_p(1)$. Thus, any arm $i \in \mathcal{I}$ must satisfy $N_i(n)/n \overset{p}{\to} 1/|\mathcal{I}|$. $\qquad \square$

# K  Proof of Fact 2 and Fact 3

## K.1  Fact 2

Since $\theta_k$ is Beta$(1, k+1)$-distributed, its PDF, say $f_k(\cdot)$, is given by

$$f_k(x) = (k+1)(1-x)^k; \quad x \in [0,1]. \tag{56}$$

Thus,

$$\begin{aligned}
\mathbb{P}\left( \theta_k > \tilde{\theta}_l \right) &= \int_0^1 \left( \int_y^1 f_k(x) dx \right) f_l(y) dy \\
&= \int_0^1 \left( \int_y^1 (k+1)(1-x)^k dx \right) (l+1)(1-y)^l dy \qquad \text{(using (56))} \\
&= \int_0^1 (l+1)(1-y)^{k+l+1} dy \\
&= \frac{l+1}{k+l+2}.
\end{aligned}$$

$\square$

## K.2  Fact 3

Since $\theta_k$ is Beta$(k + 1, 1)$-distributed, its PDF, say $f_k(\cdot)$, is given by

$$f_k(x) = (k + 1)x^k; \quad x \in [0, 1]. \tag{57}$$

Thus,

$$\begin{aligned}
\mathbb{P}\left(\theta_k > \tilde{\theta}_l\right) &= \int_0^1 \left(\int_y^1 f_k(x)dx\right) f_l(y)dy \\
&= \int_0^1 \left(\int_y^1 (k + 1)x^k dx\right) (l + 1)y^l dy \qquad \text{(using (57))} \\
&= \int_0^1 (l + 1)\left(1 - y^{k+1}\right) y^l dy \\
&= 1 - \frac{l + 1}{k + l + 2} \\
&= \frac{k + 1}{k + l + 2}.
\end{aligned}$$

$\square$