# OpenReview forum: "A Closer Look at the Worst-case Behavior of Multi-armed Bandit Algorithms"
_NeurIPS.cc/2021/Conference — NeurIPS 2021 Spotlight_

### Official Review · Reviewer_3P5t · 2021-07-16

**Rating:** 7
**Confidence:** 3

**Summary:**

Classical bandit algorithms such as UCB and Thompson Sampling are well-understood in terms of their performance—for example, an upper bound on their regret—but, until recently, little attention has been given to their actual behavior—for example, the rates at which specific arms are sampled. This work seeks to address these questions by analyzing the arm sampling distribution achieved by both UCB and Thompson Sampling. They provide results characterizing these distributions asymptotically and, in addition, characterize the distribution of the rewards obtained by running UCB.

**Limitations And Societal Impact:**

The authors do not discuss potential societal impact. However, the work is primarily theoretical so it is difficult to determine what negative societal impacts it may have. Limitations are discussed.

**Main Review:**

While the regret incurred by UCB and Thompson Sampling (TS) is well-understood, this work seeks to understand the actual behavior of these algorithms. It makes several contributions in this direction, all in the two-armed bandit problem (though some of the results extend to K arms). In particular, they characterize the:
- Asymptotic distribution of N_{I*}(n)/n achieved by UCB, where N_{I*}(n) is the number of pulls of the optimal arm up to time n, for various regimes of the gap.
- Asymptotic distribution of N_{I*}(n)/n achieved by TS but in a restricted regime where the rewards are deterministic.
- Precise regret incurred by UCB asymptotically, establishing that R_n -> c(\theta)*\sqrt{n log n} for some problem-dependent constant c(\theta).
- Distribution of the random process of the UCB’s reward and regret.

All results are asymptotic in nature (i.e. the number of samples n -> infinity). In addition, the authors provide several numerical examples illustrating that in practice UCB and TS can exhibit surprising behavior, and show that they are able to explain this behavior with their theoretical results.

Pros:
- This is a timely work. In the last few years, a significant amount of attention has been given to understanding the asymptotically optimal regret of bandit problems (e.g. [4]), as well as properties of UCB (for example, the bias in the estimates of the arm means maintained by UCB [1]). Very recently [5] studied the asymptotic sampling distribution of Thompson Sampling but, to my knowledge, no other works on this exist, though open questions still remain.
- The results are extensive and provide a full characterization of the behavior of the UCB algorithm, greatly deepening our understanding of its properties. The motivating examples help make clear that there are unusual and little-understood effects that occur when running UCB and TS.

Cons:
- The main results only hold for a two-armed bandit model, though they are partially extended in the appendix to K armed models (the restriction to two-armed bandits has precedence in the literature though, for instance [4]).
- The results on Thompson Sampling only holds for deterministic bandits. Furthermore, [5] studies the asymptotic distributions of the arm pulls of Thompson Sampling and should be cited and compared against.
- There’s a line of recent work ([1]-[3]) which aims to understand the bias of optimistic bandit algorithms. While not directly comparable to this work, it is related to the theme of understanding the behavior of bandit algorithms, and should be cited.
- Some discussion on the applications of these results would be interesting. For example, do these results motivate any improvements to UCB or other algorithms?

Overall, this is a thorough study of the performance of bandit algorithms, greatly deepens our understanding of both UCB, and warrants an accept.

----------------------------

Update after rebuttal: I would like to maintain my score after reading the rebuttal. I believe this work makes fundamental, novel contributions to our understanding of bandits and deserves an accept.


[1] Shin, Jaehyeok, Aaditya Ramdas, and Alessandro Rinaldo. "Are sample means in multi-armed bandits positively or negatively biased?." arXiv preprint arXiv:1905.11397 (2019).
[2] Shin, Jaehyeok, Aaditya Ramdas, and Alessandro Rinaldo. "On the bias, risk and consistency of sample means in multi-armed bandits." arXiv preprint arXiv:1902.00746 (2019).
[3] Shin, Jaehyeok, Aaditya Ramdas, and Alessandro Rinaldo. "On conditional versus marginal bias in multi-armed bandits." International Conference on Machine Learning. PMLR, 2020.
[4] Kaufmann, Emilie, Olivier Cappé, and Aurélien Garivier. "On the complexity of best-arm identification in multi-armed bandit models." The Journal of Machine Learning Research 17.1 (2016): 1-42.
[5] Kalkanli, Cem, and Ayfer Ozgur. "Asymptotic Convergence of Thompson Sampling." arXiv preprint arXiv:2011.03917 (2020).

**Time Spent Reviewing:**

3

---

> ### Author Response · Authors · 2021-08-10
> **Author-response**
>
> Thank you for your interest in our work and the time spent reviewing our manuscript. We greatly appreciate your feedback; point-wise responses to your main remarks are provided below.
>
> $(1)$ Extension to the K-armed setting. Although a full-scale generalization of Theorem 1 (and other results) will likely involve messy book-keeping (since $K(K-1)/2$ pairwise gaps are involved), this can be done in principle. Overall, we felt such a development is less central than other results derived in the paper given the space limits and scope.
>
> $(2)$ Comparison of Theorem 2 with suggested reference [5]. Thank you for bringing this work to our notice. While [5] indeed characterizes the asymptotic distribution of arm-pulls under Thompson Sampling, they consider the Bayesian setting where a prior distribution exists over problem instances, and information about said prior is baked into the Thompson Sampling algorithm. A sample-path of the algorithm in their model involves, in particular, a $random$ problem instance from the instance-space. In contrast, the derivation of asymptotic distribution of arm-pulls in our work is for specific $(fixed)$ problem instances, viz., reward configurations (I) and (II) described in Theorem 2, and under the classical Beta-Bernoulli version of Thompson Sampling. The two works are not comparable but are complementary, and hopefully combine to aid our understanding of Thompson Sampling. We will remark upon this in the revision.
>
> $(3)$ On suggested references [1,2,3,4]. Thank you for pointing out these works; we will include them in our literature survey as appropriate.
>
> $(4)$ Potential improvements to UCB/Thompson Sampling. As you presciently noted, the broader question posed by our work is indeed whether it is possible to design a "best of both worlds" algorithm with desirable properties of Thompson Sampling (better empirical performance in "well-separated" instances vis-`a-vis UCB) as well as that of UCB (approximately "balanced" sample-split w.h.p. when the gaps are "small/moderate," as opposed to the "imbalance" under Thompson Sampling). Such a possibility will give the user a lever to tune an appropriate operating point on the "Pareto frontier" of performance based on the desired level of trade-off between the competing objectives of regret minimization and ex post causal inference. This aspect is currently under investigation.
>
> We have noted the minor points raised as well, and will duly address them in the revision. Thank you again for all the helpful comments.

---

### Official Review · Reviewer_SFdh · 2021-07-16

**Rating:** 9
**Confidence:** 4

**Summary:**

The paper studies the asymptotical behaviors (with respect to the suboptimality gap $\Delta$) of key statistics in the standard upper confidence bound (UCB) algorithm in standard multi-armed bandits (MAB). The contributions are as follows.

1. The paper proves that the **asymptotical arm-sampling rate** converges to a constant, and gives its **analytical form** (Eq. 2). In particular, the paper discovers a non-trivial "**moderate gap**" regime for UCB.
2. The paper proves an asymptotic regret lower bound for UCB **up to $(1 + o(1))$**  in two-armed bandits.
3. The paper proves that the **asymptotical empirical sum** converges to a Brownian motion for UCB in two-armed bandits.
4. The paper proves the asymptotical arm-sampling rate for Bernoulli Thompson sampling in two-armed *deterministic* bandits.
5. The authors extend their results to MAB.

**Limitations And Societal Impact:**


### Limitations

1. The authors' notations confuse me, in that it's not clear that *which limiting regime* the authors are discussing. It seems to me that Theorem 1 is stated for a sequence of two-armed bandits instances, i.e. for each $n$, there's an instance with parameters $(\mu_{1, n}, \mu_{2, n})$ such that $\Delta_n = \mu_{1, n} - \mu_{2, n}$ satisfies $\Delta_n \asymp 1/\sqrt{n}$, and that the $N_{i^*}(n) / n$ term in the result of Theorem 1 refers to the $N_{i^*}(n) / n$ in the $n$-th instance. The confusions exist at least for Theorem 1, 3, 4. Since asymptotical style results are not common in bandits literature, I recommend the authors to point out the asymptotical regime more clearly.
2. The paper includes numerical experiments, so I think the section 3 in the checklist (Line 436) shall not be completed by "N/A", and the authors are encouraged to publish their codes, (though numerical experiments might be very simple).
3. The paper fails to fully extend the results (Theorems 1, 3, 4) to the MAB case. The two-armed case is limited from both theoretical and practical perspective. In particular, in the two-armed case, the arm pulling rates satisfy $N_1(n) = n - N_2(n)$, which makes it possible to fully characterize the bandits by only analyzing one statistics $N_1(n)$. However, in the multi-armed case, at least two statistics need be studied. The paper did not prove the most intriguing "moderate gap" part in the multi-armed case.
4. The paper's results on Thompson sampling are quite limited, in that it only studies the deterministic case where rewards are constantly $0$ or $1$ for each arm. Although the authors somehow suggest that the results should extend to the stochastic setting by doing experiments (Fig. 1), I am still concerned about the results.

### Other Comments

* Line 128: "diffuion-limit" should be "diffusion-limit"
* Line 144-145: The sentence almost repeats the same sentence at Line 137-138, same thing at Line 149-150; the authors need not repeat the claim of first characeterization three times in the same paragraph

### Societal Impact

The paper is mainly theoretical, and I don't see any potential negative societal impact.

**Main Review:**


While asymptotical analyses are common in statisics and EE (e.g. mean-field asymptotics), it is not common in CS and rare in bandits literature. Asympotical analyses are important because it could help both researchers and practitioners better understand the algorithmic behaviors, because non-asymptotic analyses are usually not as exact and precise as their asymptotical counterparts.

This paper studies two-armed bandits, the classical and fundamental task in bandits literature, and analyzes UCB, the classical and fundamental algorithm in bandits literature. The most important statistics maintained by the UCB algorithm are the number of times of each arm being pulled and the empirical sum of each arm. For these two statistics, the paper provides exact asymptotical characterization under the simplest two-armed bandits case, which clearly distinguishes this paper from previous papers. The results are significant and set up seminal directions in bandits literature, and could benefit subsequent researchers, including myself. I believe the related works are cited adequately.


The paper is very well-written. The maths are easy to read and can be clearly verified in a step-by-step manner. I went through Appendices E and F (which contains main technical proofs) and did not find apparent errors. The authors are honest about their weakness about their results in the final section.

The paper almost completely depicts how standard UCB works in two-armed bandits. The results on other tasks such as multi-armed bandits and other algorithms such as Thompson sampling are, however, still premature. Further will be discussed in the limitations section. Nonetheless, given the novelty, quality, and clarity of this submission, I vote for a strong accept.

**Time Spent Reviewing:**

7

---

> ### Author Response · Authors · 2021-08-10
> **Author-response**
>
> Thank you for your interest in our work and the time spent reviewing our manuscript. We greatly appreciate your feedback; point-wise responses to your main remarks are provided below.
>
> $(1)$ Clarification on notation. We regret the confusion our notation might have caused. Indeed, your interpretation in terms of a sequence of two-armed bandit instances indexed by $n$ (the horizon of play) is correct. We will make the notation unambiguous in the revision; thank you for pointing this out.
>
> $(2)$ On publication of codes. We have noted this and will certainly do the needful; thank you for pointing this out.
>
> $(3)$ Generalization to the $K$-armed case. Although a full-scale generalization of Theorem 1 (and other results) to K-MAB will likely involve messy book-keeping (since $K(K-1)/2$ pairwise gaps are involved), this can be done in principle (A simple extension of Theorem 1 to the $K$-armed setting is provided in Appendix B for illustrative purposes.). Overall, we felt such a development is less central than other results derived in the paper given the space limits and scope.
>
> $(4)$ Results for Thompson Sampling. This is still an evolving landscape. A recent paper [Lin Fan and Peter W Glynn. Diffusion approximations for Thompson sampling. arXiv preprint arXiv:2105.09232, 2021] appeared post our submission; cited paper derives a diffusion limit for Thompson Sampling in the Gaussian bandit setting (distinct from our Beta-Bernoulli setting) and does so using a very different framework that is not directly applicable to our setting. Moreover, unlike the closed-form limit for UCB (Theorem 4 in our paper), the limit-process for Thompson Sampling can only be characterized in terms of solutions (possibly non-unique) to a stochastic ordinary differential equation driven by a time-changed Brownian motion. The complex nature of the limit has to do with the non-degeneracy of the distribution of $N_i(n)/n$ under Thompson Sampling as $n\to\infty$, when $\Delta \asymp 1/\sqrt{n}$; a special case of this phenomenon is the "imperfect learning" observable in instances with "zero gap," which Theorem 2 in our paper formalizes in the deterministic setting. It is readily observable empirically that non-degeneracy in the asymptotic distribution of $N_i(n)/n$ persists up to diffusion-scale $\mathcal{O}\left( 1/\sqrt{n} \right)$ gaps under Thompson Sampling, both in the Beta-Bernoulli as well as the Gaussian setting (see [25] for a few examples in the Gaussian setting). Theoretical development in this area is still in its initial stages at the moment, and we hope Theorem 2 provides a useful starting iterate for further investigations into this aspect of Thompson Sampling.
>
> We have noted the minor points raised as well, and will duly address them in the revision. Thank you again for the careful reading.

---

### Official Review · Reviewer_KzpM · 2021-07-16

**Rating:** 7
**Confidence:** 3

**Summary:**

The authors study the asymptotic behavior of arm-sampling distributions under the UCB and the Thompson sampling. They provide an asymptotic characterization of the distributions, and show the arm sampling rates asymptotically deterministic regardless of the hardness of instances. With this characterization, focused on canonical UCB algorithm, they provides the first algorithm-specific worst case bound and the first diffusion-limit performance.


**Limitations And Societal Impact:**

A limitation of that paper is to deal with two-armed bandit settings, but as described in the concluding remarks in the paper, it may be generalized K-armed bandit. So, I think the contributions outweigh this drawback.

I found the following researches that also consider the asymptotic regime using a diffusion approximation. It would be better to add some if possible.

References:

[i] Keisuke Hirano and Jack R Porter. Asymptotic representations for sequential experiments. Cowles Foundation Conference on Econometrics, 2021

[ii] Fan, Lin, and Peter W. Glynn. "Diffusion Approximations for Thompson Sampling." arXiv preprint arXiv:2105.09232 (2021).

[iii] Araman, Victor F., and Rene Caldentey. "Diffusion Approximations for a Class of Sequential Testing Problems." arXiv preprint arXiv:2102.07030 (2021).


**Main Review:**


Originality:
To the best of my knowledge, this is the first work to provide algorithm-specific result (Theorem 3). Also, it is interesting that they provide another view of proving the worst-case performance of UCB using a diffusion scaling approach (Theorem 4). Although the paper is not the first work introducing the diffusion scaling to bandit problems, it can deal with more standard bandit algorithms. As for the originality of proof of Theorem 4, proof techniques seem to be similar to that of [25]. Is there any difficulty to apply a diffusion scaling technique to more standard bandit algorithms?

Quality:
The proposed theorems seem to be technically sound. The authors honestly discuss about the strengths and weaknesses.

Clarity:
The paper is well-written and provides appropriate examples. I can easily follow the motivation. Since the diffusion scaling is a key technique of the main contribution, more descriptions about it helps us to understand the details of the contributions.

Significance:
As written in the originality, the provided results are the first works, which is the significant point of the paper. Since the works include some limitations that deals with two-armed bandit settings, it is an important first step to explore other bandit algorithms and settings .


**Time Spent Reviewing:**

6 hours

---

> ### Author Response · Authors · 2021-08-10
> **Author-response**
>
> Thank you for your interest in our work and the time spent reviewing our manuscript. We greatly appreciate your feedback; point-wise responses to your main remarks are provided below.
>
> $(i)$ On the connection between Theorem 4 and the derivation in [25]. The distinctions have been discussed in Lines 317-320, and again in Lines 344-354 in the paper. In a nutshell, [25] uses the martingale framework developed by [Daniel W Stroock and SR Srinivasa Varadhan. Multidimensional diffusion processes. Springer, 2007] to characterize the diffusion limit for algorithms satisfying certain regularity conditions, whereas we derive Theorem 4 for UCB [6] directly from first principles. More importantly, our work is independent of [25] as the latter's framework cannot be applied to UCB due to a violation of the aforementioned regularity conditions under UCB. In fact, the framework of [25] is quite limited insofar as its applicability to the study of bandit algorithms; in addition to  UCB, both versions of Thompson Sampling discussed in [2] also remain outside the ambit of the analysis in [25].
>
> $(ii)$ On the difficulty of deriving diffusion approximations for general algorithms. As noted in the paper, the fact that $N_i(n)/n \xrightarrow{p} 1/2$ for both arms $i=1,2$ under UCB when $\Delta \asymp 1/\sqrt{n}$, is crucial for the diffusion-limit process to admit a closed-form characterization as stated in Theorem 4. Theorem 2 shows that the aforementioned condition $\left(N_i(n)/n \xrightarrow{p} 1/2\right)$ is provably violated under Thompson Sampling in the Beta-Bernoulli bandit setting. In fact, similar empirical observations are available also in the Gaussian bandit setting. The [Fan and Glynn] reference you mention, derives a diffusion-limit process for Thompson Sampling in the Gaussian bandit setting using a very different framework that is not directly applicable to our setting. Moreover, unlike our result for UCB, the limit-process in their work can only be characterized in terms of solutions (possibly non-unique) to a stochastic ordinary differential equation driven by a time-changed Brownian motion. This characterization has to do with the non-deterministic nature of $N_i(n)/n$ under Thompson Sampling when $\Delta \asymp 1/\sqrt{n}$. Presently, we are unaware of a general strategy for deriving diffusion approximations for bandit algorithms. Our approach is tailored to UCB (and UCB-like) algorithms under which $N_i(n)/n \xrightarrow{p} c$ when $\Delta \asymp 1/\sqrt{n}$, where $c$ is some constant bounded away from $0$ and $1$. The work in [Fan and Glynn] is specific to Thompson Sampling in the Gaussian bandit setting. As discussed earlier, [25] proposes a general framework to derive diffusion approximations. However, their conditions are restrictive and the classical versions of UCB as well as Thompson Sampling remain outside the ambit of their analysis.
>
> $(iii)$ Miscellaneous. A simple extension of Theorem 1 to the $K$-armed setting is  provided in Appendix B. A full-scale generalization involves messy book-keeping since $K(K-1)/2$ pairwise gaps are involved, but this can be done in principle (though we felt it is outside the scope of the present conference paper).  Of the three references suggested, [Fan and Glynn] is certainly relevant to our work; thank you for bringing this to our notice. [Hirano and Porter] and [Araman and Caldentey] study diffusion limits in broader settings involving sequential experiments (which are quite distinct in nature from our bandit paradigm); the intersection with these papers is minor, and it is worth noting that their analysis does not directly pertain to analysis of UCB and similar bandit algorithms.
>
> We have noted the minor points raised as well, and will duly address them in the revision. Thank you again for the helpful comments.

---

### Official Review · Reviewer_15pu · 2021-07-16

**Rating:** 7
**Confidence:** 4

**Summary:**

The paper studies the arm sampling behavior of UCB and Thompson sampling algorithms. For the two-arm case, the asymptotic behavior of arm sampling is characterized for different regimes (small, large, and medium) of suboptimality gap. Using this characterization, the minimax regret of UCB is shown to O(n\logn), where n is the time horizon. They highlight the incomplete learning phenomenon in Thomson sampling where sample-split could be arbitrarily imbalanced along a sample path even when both the arms have the same mean.

**Limitations And Societal Impact:**

Yes

**Main Review:**

The paper highlights the important aspect of arm sampling distribution in the Multi-armed setting. The author nicely built the imbalances in arm sampling through examples for both UCB and Thompson sampling. The insights from the analysis highlight the 'incomplete learning' learning aspect in Thompson sampling.

I have the following question for the authors:

Theorem 1: The result holds aysmptotically and $\Delta$ (problem instance) is changing with $n$. After $n$ rounds (sufficiently large), we expect UCB to have played the optimal arm exponentially more time than the sub-optimal arm. But the asymptotic result suggests that this need not be the case if $\Delta \rightarrow 0$ as $n \rightarrow$. Can anything be said about the arm-sampling distribution for a fixed instance $\Delta$. The asymptotic result is good, but since we also know finite time behavior, it is nice to connect.

Theorem 2: What does the symbol $\implies$ denote (convergence?). The statement assumes $n\rightarrow \infty$, but the Proof Sketch says for any $n \in \mathcal{N}$, which one is correct.

Theorem 3: Does the result hold with weaker condition $\Delta \rightarrow 0$ Instead of thanking $\Delta \sim \sqrt{\frac{\theta \log n}{n}}$.

Overall nice paper. Good insights. If authors could highlight the practical implications of their results, it could be nice. For example, is incomplete learning of Thompson Sampling good or bad. When should UCB be preferred over Thompson sampling and vice versa?
Does the incomplete learning aspect explain why Thompson sampling often performs better empirically in practice?

I would be happy to revise the scores post rebuttal.
-----

Post rebuttal:

Thanks for clarifying some of my points. I have increased the score to 7.


**Time Spent Reviewing:**

8

---

> ### Author Response · Authors · 2021-08-10
> **Author-response**
>
> Thank you for your interest in our work and the time spent reviewing our manuscript. We greatly appreciate your feedback; point-wise responses to your main remarks are provided below.
>
> $(i)$ $Theorem1.$ Finite-time results are indeed well-known for UCB. However, the assertion that the optimal arm is played exponentially more often than the sub-optimal ones is true only when the instance is "well-separated." In the context of the two-armed problem we study, a "well-separated" instance corresponds to a fixed value of $\Delta$ that is bounded away from $0$ and independent of the horizon of play $n$ (basically, $\Delta$ being some $positive$ absolute constant). Such an instance is an element of the "large gap" regime where $\Delta = \omega \left( \sqrt{{\log n}/{n}} \right)$. In the "large gap" regime, the finite-time behavior of the right tail of the distribution of $N_i(n)/n$ ($i$ being the inferior arm among the two) under UCB is well-documented in [5] (see the the literature review section of the main paper at Line 120). However, the results in [5] become vacuous if $\Delta$ scales with $n$ and vanishes at $\mathcal{O} \left( \sqrt{{\log n}/{n}} \right)$ rates. One of the technical contributions of our paper is deriving finite-time tail bounds for $N_i(n)/n$ in the $\Delta = \mathcal{O} \left( \sqrt{{\log n}/{n}} \right)$ regime. We derive and use such bounds in the proof of Theorem~1, but the details are currently included only in the supplementary material due to space constraints; we will revise the exposition of the main paper based on your suggestion.
>
> $(ii)$ $Theorem2.$ '$\implies$' denotes weak convergence, as noted in Line 175. The proof sketch remarks that in the $q=1$ case, $N_1(n)$ is uniformly distributed over $\{0,1,...,n\}$ for any $n\in\mathbb{N}$. The assertion in part (II) of the theorem follows as a corollary to the aforementioned statement; to the best of our understanding there is no contradiction here. The full technical details can be found in Appendix~G.
>
> $(iii)$ $Theorem3.$ The result as stated is specific to the "moderate gap" regime where $\Delta  \asymp \sqrt{{\log n}/{n}}$. The proof sketch of Theorem~3 (Line 298-305) remarks that the limit of ${R_n^\pi}/{\sqrt{n\log n}}$ is non-trivial only in the "moderate gap" regime. In other regimes of $\Delta$, viz., $\omega\left( \sqrt{{\log n}/{n}} \right)$ "large" and $o\left( \sqrt{{\log n}/{n}} \right)$ "small" gaps, the limit is $0$.
>
> $(iv)$ $Miscellaneous.$ We will place a greater emphasis on the practical implications of the phenomena of "incomplete learning" under Thompson Sampling and "balanced sample-split" under UCB; thank you for the suggestion. On your last point regarding "incomplete learning," we do indeed believe that it is strongly tied to Thompson Sampling's better empirical (regret) performance vis-\`a-vis UCB. Having said that, it seems far from obvious since the state-of-the-art minimax regret bounds in prior work are $\mathcal{O}\left( \sqrt{n\log n} \right)$ for both algorithms. For UCB, our present work improves upon the state-of-the-art by showing an $\Omega \left( \sqrt{n\log n} \right)$ lower bound; thus the minimax regret of UCB is, in fact, $\Theta\left( \sqrt{n\log n} \right)$. For the version of Thompson Sampling studied in this paper (Beta priors and Bernoulli likelihoods), unfortunately, no lower bound on the minimax regret is currently known in the literature. As a result, despite compelling empirical evidence, the claim that Thompson Sampling incurs a smaller regret than UCB is still not theoretically supported. However, it is quite likely that the claim is true, considering well-known related results for the minimax regret of Thompson Sampling with Gaussian priors and Gaussian likelihoods, which is known to be $\Theta\left(\sqrt{n}\right)$ [2]. Furthermore, "incomplete learning" is also observable empirically under said algorithm. The same is true also for other minimax-optimal algorithms such as MOSS [Audibert and Bubeck, Minimax policies for adversarial and stochastic bandits, COLT 2009]; a minimax regret of $\Theta\left(\sqrt{n}\right)$ and an empirically observable "incomplete learning" phenomenon. We do believe based on these observations that the "incomplete learning" phenomenon for an algorithm is somehow connected to its minimax optimality. This also suggests that it may very well be possible to "shave off" the $\sqrt{\log n}$ term from the current state-of-the-art $\mathcal{O}\left( \sqrt{n\log n} \right)$ bound for the Beta-Bernoulli version of Thompson Sampling studied in this paper. The theoretical development in this space, however, is in a very nascent stage at the moment, and much remains to be done to iron out these conjectures. We hope the present work helps build some foundations towards these future investigations.
>
> We have noted other minor points raised as well, and will duly address them in the revision. Thank you again for the many helpful suggestions.

---

### Decision · Program_Chairs · 2021-09-27

**Decision:**

Accept (Spotlight)

**Comment:**

This is strong paper that I would like to see accepted as a spotlight. There is a minor criticism concerning the restriction to two arms but the reviewers feel that this is not a major concern. There are some suggestions by the reviewers of how to improve the paper further and it would be good if the authors could incorporate these.